# Classification and Validation of Spatio-Temporal Changes in Land Use/Land Cover and Land Surface Temperature of Multitemporal Images

**Vimala Kiranmai Ayyala Somayajula [1], Deepika Ghai [1,*], Sandeep Kumar [2], Suman Lata Tripathi [1], Chaman Verma [3], Calin Ovidiu Safirescu [4,*] and Traian Candin Mihaltan [5]**

[1] School of Electronics & Electrical Engineering (SEEE), Lovely Professional University, Phagwara 144411, India
[2] Koneru Lakshmaiah Educational Foundation, Vaddeswaram 522302, India
[3] Department of Media and Educational Informatics, Faculty of Informatics, Eötvös Loránd University, 1053 Budapest, Hungary
[4] Environment Protection Department, Faculty of Agriculture, University of Agricultural Sciences and Veterinary Medicine of Cluj-Napoca, Calea Mănăştur No. 3–5, 400 372 Cluj-Napoca, Romania
[5] Faculty of Building Services, Technical University of Cluj-Napoca, 400 114 Cluj-Napoca, Romania
* Correspondence: deepika.21507@lpu.co.in (D.G.); calin.safirescu@usamvcluj.ro (C.O.S.); Tel.: +91-9463647809 (D.G.)

**Abstract:** Land transfiguration is caused by natural as well as phylogenesis-driving forces, and its consequences for the regional environment are a significant issue in understanding the relationship between society and the environment. Land use/land cover plays a crucial part in the determination, preparation, and execution of administrative approaches to fulfilling basic human needs in the present day. In this study, Visakhapatnam, Vijayawada, Tirupati, A.P., India, is considered as a study area to explain the Land use/land cover (LULC) classification, Land Surface Temperature (LST), and the inverse correlation between LST and the NDVI of Temporal Landsat satellite images at intervals of 5 years from 2000 to 2020. We performed easy and thoroughgoing classifications based on vegetation phenology, using an extended LULC field database, a time series of LANDSAT satellite imagery, and a pixel-based classifier. In total, five land-use and land-cover types have been identified: dense vegetation, vegetation, built-up, barren land, and water. Over the period of inquiry, there were notable increases in the area of built-up land, dense vegetation, and vegetation, whereas there was a marked decrease in water bodies and barren land. The diverse effects of land transformation on the natural environment have been assessed using Land Surface Temperature (LST) and the Normalized Difference Vegetation Index (NDVI). The used technique achieved very good levels of accuracy (90–97%) and a strong kappa coefficient (0.89–0.96), with low commission and omission errors. The variation of the land surface temperature was studied using the Mono-Window algorithm. Change detection, and the transition of the natural land cover to man-made land use, were statistically computed for the study area. Results exposed that there had been significant variations in the land use and cover during the tagged eras. In general, two land use and land cover change patterns were confirmed in the study zone: (i) compatible growth of the zone in built-up areas, barren land, plantations, and shrubs; and (ii) continual diminishment in agriculture and water; maximum urban development took place between 2000 to 2020. The results showed drastic changes in urbanization and decrements in vegetation that had environmental consequences.

**Keywords:** land use/land cover; remote sensing; NDVI; PCA; MWA; OLI; TIRS; LST

## 1. Introduction

Land-use/land-cover (LULC) modifications are significant issues of worldwide ecological change. With their abundant nature, satellite remote sensing data have been valuable in mapping LULC patterns over time. The LULC of a region results from the conventional economic needs of humanity in reality. The land is becoming a scarce asset because of massive

rural and segment pressure. In the present situation, LU and LC classification using remote sensing images is an essential part of several applications like the management of biological resources, agricultural practices, land-use planning, and forest management [1–3]. It is necessary to cognize LULC information to understand the current status of the land, plan for the future based on climatic changes, and assure the feasibility of accessible resources [4,5]. Land cover is related to how the Earth's exterior is defended by timberlands, marshes, impenetrable shells, civilization, and water [6,7]. Land use provides information about how mankind uses the topography for improvement and sustainability. Human activities will influence the environmental condition of the future, as the population increases exponentially. Sprawl, unintended decolonization, and related rapid population growth have apparent calamitous effects on territorial environs [8]. LULC evaluation is essential in solving environmental disputes such as unstructured evolutions, loss of cultivation lands, and the destruction of marshland at the territorial, national, and global levels [9,10]. LULC modification surveys help in explicating: (i) which consequence is taking place, (ii) which type of land cover is replacing the previous type, (iii) the form of alteration taking place, (iv) the rate of land change, and (v) the ultimate reason for the change [11,12].

The quick expansion of administrative divisions globally has become an environmental problem in the 21st century, and it is one that requires the acceptance of fresh scientific formulations and new sources of data [13,14]. These sudden environmental alterations at local, territorial, and worldwide levels have constituted earnest warnings to humans [15]. Therefore, an aggregate of data from remote sensing (RS) and geographic information system (GIS) applications, with preferred spatial and spectral resolutions, provides the information wished for in environmental effect analysis. Rahman et al. [16] (2020) concentrated on analyzing the LULC change pattern in Bangladesh from 1990 to 2019 using the maximum likelihood (ML) classifier, and they discovered that prime consequences take place in urban areas, timberland, water bodies, and vegetation-covered areas. The assessment of LULC change is a distinguishing proof of a fruitful arrangement, and provides broad approaches to studying the geographic area, land-use procedures, and geophysical science structures in the survey area [17].

Multitemporal and multispectral high- and moderate-spatial-resolution satellite data act as key tools for approximating features such as vegetation, forest degradation, and an increase in urbanization [18]. The different environmental influences of LULC have been analyzed in different ways, comprising remote sensing classifications of surface alterations using LAI, LST, NDVI, and NDWI [19–22]. LST is frequently used to obtain global temperature alteration and geophysics, and geo-biophysical and LULC research [23–26]. Here, LST, NDVI, and NDWI were utilized. Remote sensing is a phenomenon used in observing or exploiting information about distant objects without physical contact and in contrast to in situ or on-site observation.

Remote sensing is "the science and art of prevailing information about an object, region or process through the analysis of data acquired by a device that is not in contact with the object, region or process under investigation" [27]. Remote sensing (RS) is, in particular, mentioned as the build-up and illustration of information about a physical item or area without being in physical touch with the item or area [28]. Remote sensing can be utilized to evaluate the earth's conditions over a very large area, and it also allows for the observation of changes in the environment. Multispectral images are applied to determine the NDVI, which is an essential amount within side the evaluation of land cover/use categorization. NDVI is primarily based on divergence among the red band and near-infrared band of satellite images. The range of NDVI is from −1 to +1. NDVI is exploited to indicate the inexperienced flora index which includes quantities of chlorophyll, inexperienced flora ease, and leaf area index [29–31].

Landsat OLI pictures (2013 and 2015) were chosen in separating the LULC data of Nyingchi Country [32]; the DEM was utilized to extricate articles' property bent surface region and break down their three-layered powerful change data, which acknowledged four-layered observation of the ranger service data on the schedule and spatial levels. In

formal procedures, mapping is executed with the help of reachable records, field studies, and maps. Therefore, formal procedures are time-consuming and costly. Furthermore, the created maps quickly transform noncurrent factors in the dynamic environment [33,34]. LULC changes in large cities can be obtained with the help of aerial photos and high-resolution images, but a limited database is available because of commercial enterprise constituents. Still, medium-resolution information such as Landsat5, Landsat8 OLI images, Multi-Spectral Scanner (MSS), Thematic Mapper, and ETM+ is available globally in LULC change detection [35,36]. Vani et al. [5] (2020) made use of TM, ETM+, and OLI images to represent urban development-related land surface temperature changes in Vijayawada city of Andhra Pradesh state, India.

Rahaman et al. [2] (2020) appraised the consequences of LULC change on the environs of Bardhhaman district, West Bengal, using TM and OLI/TIRS images of Landsat8. A geographic information system (GIS) is a computing device used to explore and present geographically recommended information. It uses data that are attached to a unique location. Arveti et al. [37] (2020) represented the LULC status of the Tirupati region, Andhra Pradesh exploitation a unified conceptualization of RS and GIS illustrated an important effect of sprawl on the ecosphere. Abasement caused by tourist expansion, and the conversion of cultivated land via colonization results in profound changes in the environment. This consequence badly affects the local/territorial/worldwide environment. Urban lifestyles support a judgment that 60% of the global population will be urbanized by 2025 (United Nations Population Fund (UNPF)) [38]. Sundara et al. [39] (2012) applied the maximum likelihood classifier and obtained LULC changes and urban sprawl assessments of Bezawada city of the years 1990 and 2009: overall accuracy and kappa coefficient were given as 86.67%, 0.8 (1990), 85%, 0.78 (2009), respectively.

Yerrakula et al. [40] (2014) analyzed urban sprawl changes and detected LULC changes in Bezawada city using the minimum distance classifier and got overall accuracy, kappa coefficient as 67.19%, 0.6405. Vani et al. [5] (2020) assessed spatio-temporal consequences in LULC, urban sprawl, and LST throughout Vijayawada city in the years 1990, 2000, 2010, 2018 with the help of ML Classifier and NDVI and found overall accuracy of 96.33%, 93.07%, 92%, and 87% with kappa coefficients as 0.938, 0.869, 0.88, and 0.806. Rao et al. [41] (2018) compared 2013, and 2014 data by applying various classification techniques like parallelepiped (OA—82.426 kappa—0.804 for the year 2013, OA—94.167 kappa—0.919 for the year 2014), minimum distance classifier (OA—88.808, kappa—85 for the year 2013 and OA—84.028, kappa—0.794 for the year 2014), Mahalanobis classifier (OA—89.055, kappa—0.842 for the year 2013 and OA-89.278, kappa—0.857-for the year 2014), maximum likelihood classifier (OA—90, kappa—0.899 for the year 2013 and OA—91.194, kappa—0.906 for the year 2014).

In the previous studies, many researchers used minimum distance classifier, maximum likelihood (ML) classifier and parallelpiped classifier to classify land-cover changes. The disadvantages of minimum distance classifier are listed as follows:

(i) In the minimum distance method if any unclassified pixels are present, the algorithm of the minimum distance gets slightly more complicated. Another issue with the minimum distance classifier algorithm is that there will be misclassification when all pixels are classified, even if the shortest distance is far away.

(ii) Accuracy is low compared to other methods, such as ML classifier and Mahalanobis classifier.

(iii) It is time-consuming to count samples, but there is a need for more samples for high accuracy. Thus, there is a trade-off between accuracy and time complexity. If the number of samples increases accuracy, it does so at the cost of time complexity.

The disadvantages of the maximum likelihood (ML) classifier are listed as follows:

(i) Adequate ground truth data should be sampled to confirm the assessment of the mean vector and the variance–covariance matrix of the population.

(ii)   The inverse matrix of the variance–covariance matrix turns out to be unbalanced in the case where there is a very high correlation between two bands, or the ground truth data are very homogeneous.

(iii)  The maximum likelihood technique cannot be functional when the dispersal of the population does not follow the normal distribution.

In a parallelpiped classifier, the accuracy will be low especially when the distribution in the feature space has covariance or dependency with the oblique axis. To overcome these problems, the interactive supervised classification technique is used to bring out the supervised class without developing a signature file. It increases the speed of the categorization.

*Motivation behind the Present Study*

Vijayawada of Andhra Pradesh state (AP) is one of the pedagogic centers of AP, and here one can observe the speedy population migration and sprawl of the city. Over the past few decades, the city extended at the price of normal foliage, hills, and water bodies, causing the city to lose green spots and wetlands. As Visakhapatnam is a tourist place in Andhra Pradesh state, many changes are made to improve the city's economic status. In this direction, the main objective is

(i)    To analyze LULC changes in Vijayawada, Visakhapatnam, and, Tirupati and assess spatio-temporal variations based on LULC changes.

(ii)   To analyze the transformation of surface temperature between vegetated and urbanized areas, correlating 20 years of considered data from LST, NDVI, and associated with the possible seasonal influences. The study shows the relationship between LST and NDVI during a rapid urbanization process, and how land use and land cover changes can affect this relationship.

## 2. Study Area

**Vijayawada (VJY)** is an ancient city located in the middle of A.P., and is the second-biggest metropolis in AP. The metropolis reclines at the slope of River Krishna, enclosed with the aid of using the hill of the Eastern Ghats famed as Indrakeeladri Hills in Krishna District. The metropolis is mulled over as a hallowed spot for a home, holding the highly visited and famous sanctuaries of Andhra Pradesh, India, Kanaka Durga Temple of Hindu Goddess Durga. The metropolitan network is geologically mendacious in Andhra Pradesh alongside the banks of Krishna River with a scope of sixteen 003′11″ N and longitude eighty 00 3′91″ E (e.g., Harika, 2012). It is a metropolitan network with historical, governmental, educational, and social backgrounds. The environmental fame is equatorial, with fiery summertime seasons and mild winters. The elevated warmth reaches 47 °C in the May–June period, and conversely is far down around 20–270 °C in the winter months. The moistness is 78%, and the yearly downpour is 103 cm. Figure 1 presents the geographical location of Vijayawada city [42–44].

**Visakhapatnam (VSP)** additionally referred to as Vizagapatam, Vizag, or Wāltair is the executive running capital of Andhra Pradesh state. It is moreover the most occupied and vital town in Andhra Pradesh. It lies among the Eastern Ghats and the coastline of the Bay of Bengal. It is the second largest town on India's east coast after Chennai and the fourth largest town in South India, as is addressed in Figure 2. It is one of the four clever cities of Andhra Pradesh assigned under the Smart Cities Mission (The Times of India). The town is arranged to lie along 17.7041 N and 83.2977 E. (www.fallingrain.com, Seta et al., 2016) [45,46]. Visakhapatnam has an equatorial, marshy and arid environmental condition. Every year suggests temperatures guaranteed to be between 24.7–30.6 °C (76–87 °F), with the extremum in May and the nominal in January; the nominal temperatures vary from 20–27 °C. The extremum temperature registered became 42.0 °C, and the least became 20.0 °C. It has rainfall from the southwest and northeast monsoons and suggests every year rainfall registered is 1200 cm.

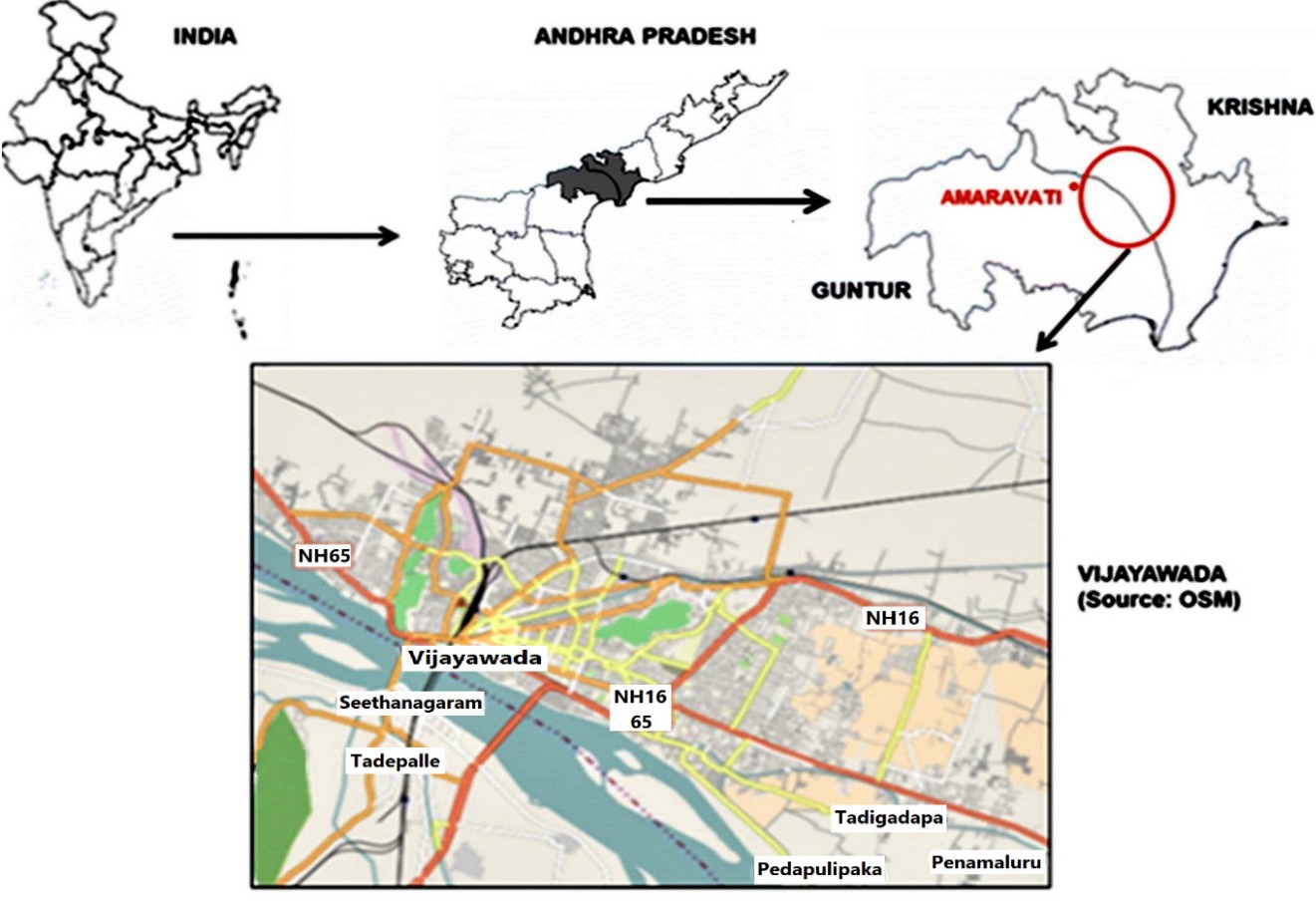

**Figure 1.** Geographical location of Vijayawada city.

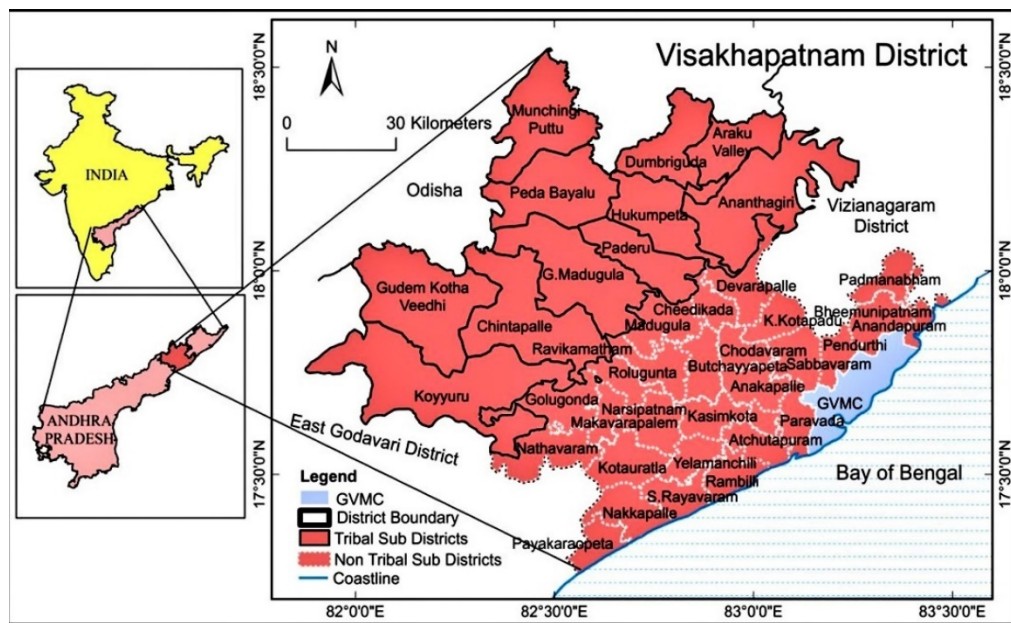

**Figure 2.** Geographical location of Visakhapatnam district.

**Tirupati (TPT)** is placed at 13.65° N 79.42° E in the Chittoor District A.P. It lies at the lowest point of the Seshachalam Hills of the Eastern Ghats, which were malleable throughout the Precambrian era. Situated 750 km southwest of the state's government capital

of Visakhapatnam, the town is in the vicinity of the important Hindu shrine of Tirumala Venkateswara Temple and several different ancient temples, being cited as the "Spiritual Capital of Andhra Pradesh", as is confirmed in Figure 3. Tirupati has an equatorial marshy and arid environmental situation decided on below the Köppen environmental situation. In wintertime, the marginal temperatures are betwixt 18 °C and 20 °C (64.4 and 68.0 °F). Unremarkably, summertime lasts from March to June, with the advent of a shower period in July, preceded by the winter, which stays until the quiet of February. The town encounters dense rainfall in November throughout the northeast monsoon season.

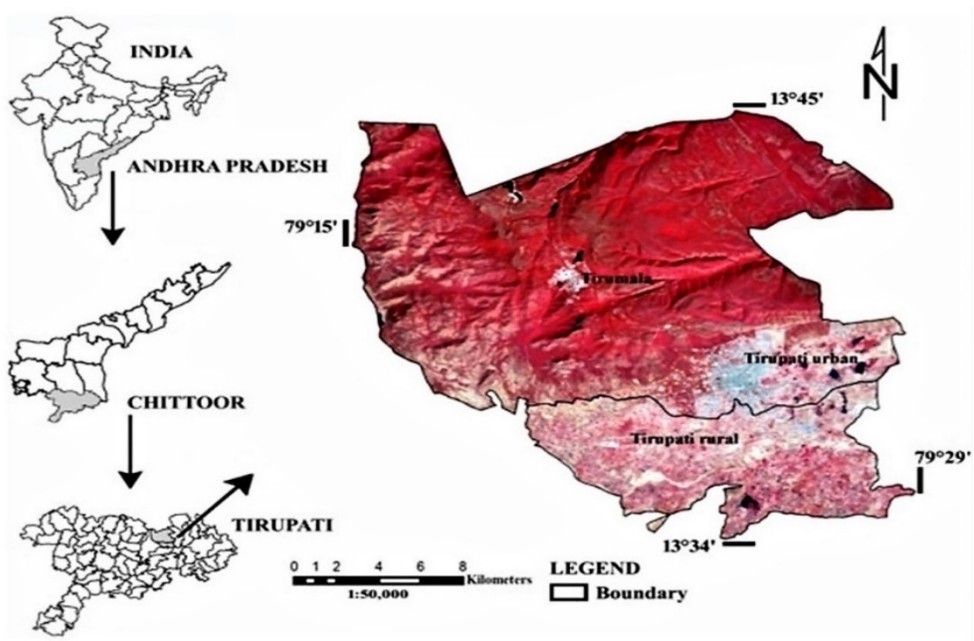

**Figure 3.** Geographical location of Tirupati.

### 3. Methodology

*3.1. Establishment of Temporal LULC Maps of the Survey Region*

This survey generally concentrated on rendering the consequences of land use via satellite imagery and statistical information. The numerical mechanisms of alternate detection are exploited in this research. In the alternate detection mechanism, each satellite image is categorized. The resultant LULC maps acquired after the type are analogized to the pixel-by-pixel conceptualization with the aid of using a change detection matrix.

**Step 1—Data Collection:** Time collection Landsat satellite information (Thematic Mapper—TM and Operational Land Imager—OLI) had been exploited to originate the LULC maps of Vijayawada, Visakhapatnam, and Tirupati for the years 2000 (TM—27 April), 2005 (TM—26 April), 2010 (TM—9 May) 2015 (OLI—28 April) and 2020 (OLI—9 April) using red, and near-infrared (NIR) bands of 30 m resolution shown in Figure 4, which represents the flowchart of the proposed method. Spatial information of the study area is represented in Table 1. The worldwide reference system (WRS) is a worldwide documentation framework for Landsat information. WRS empowers a consumer to enquire about satellite imagery of any section of the universe by determining an apparent aspect of place allotted by path and row numbers. The accumulation of a path and row number unambiguously verified a specified aspect center. The path number is given first, followed all of the time by the row number. The images were downloaded using USGS Earth Explorer (http://earthexplorer.usgs.gov/ (accessed on 19 October 2022)).

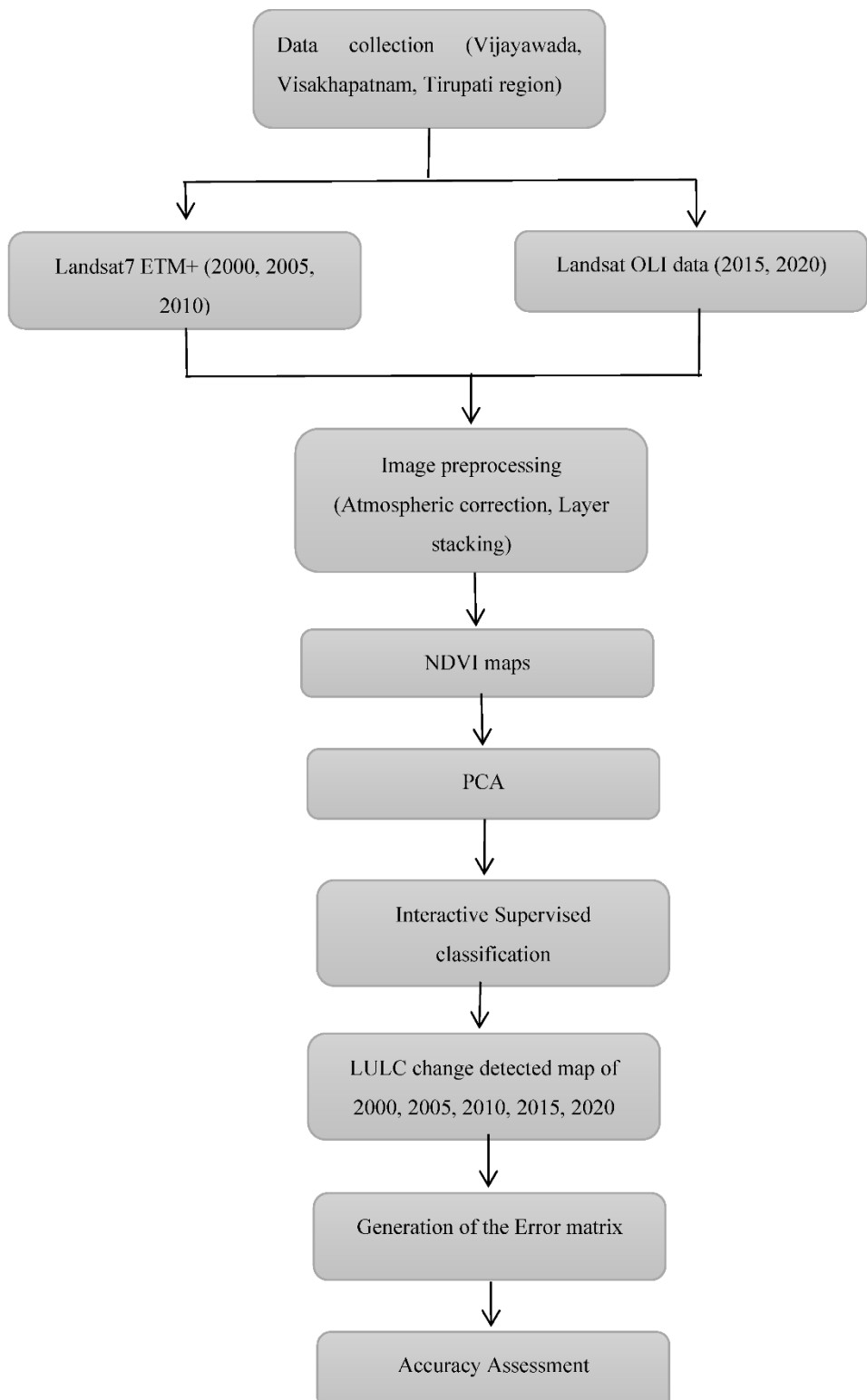

**Figure 4.** Flowchart for generation of temporal LULC maps of the survey region.

**Table 1.** Spatial information of the study area.

| Area | Data Source | Sensor | PATH | ROW |
|---|---|---|---|---|
| Visakhapatnam | Landsat | ETM+, OLI/TIRS | 43 | 33 |
| Vijayawada | Landsat | ETM+, OLI/TIRS | 25 | 36 |
| Tirupati | Landsat | ETM+, OLI/TIRS | 96 | 78 |

**Step-2—Image Pre-processing:** Radiometric correction is an error that causes the radiance or radiometric value of a scene element. Radiometric correction is performed to reduce errors in digital numbers of images and improves interpretability and analysis, to standardize images.

The initial step is to transform digital number (DN), i.e., pixels in an image, into top of atmosphere (TOA) radiance. For Landsat images, spectral radiance at the sensor's aperture is given by the equation (Equation (1)).

$$\text{DN values to TOA reflectance} = \text{band specific reflectance\_multi\_band} \times \text{DN values} + \text{reflectance\_add\_band} \tag{1}$$

The correction of the Sun angle is further performed using the equation (Equation (2)).

$$\text{Correction for Sun angle} = \text{TOA reflectance}/\sin(\text{sun elevation}) \tag{2}$$

For an accumulation of various origin data in a classification procedure, radiometric and geometric corrections play a vital role which was obtained using image pre-processing techniques [42,47]. Multiple bands were combined into a single image using layer stacking.

**Step-3—NDVI maps:** NDVI is a generally used vegetation index obtained from RS assets, gauging photosynthetic radiation consumed by Earth's surface. NDVI maps were generated using the OLI sensor's Band4 (RED), and Band5 (NIR). The NDVI values ranges from −1 to +1, with positive values denoting dense vegetation and negative indicating water [48] and are given by the equation (Equation (3)).

$$\text{NDVI} = \frac{\text{NIR} - \text{RED}}{\text{NIR} + \text{RED}} \tag{3}$$

**Step-4—Principal Component Analysis:** principal component analysis (PCA) was applied on generated NDVI maps, a mechanism for renewing raw data into a new set of information that can provide the analytical content. A set of correlated primary bands are reinforced towards a distinct uncorrelated variable, comprising initial progressive data that must be surveyed [49,50].

The steps to calculate PCA are listed as follows:

1. Calculate the mean of the image matrix, the mean vector being the vector average of the individual components of a vector (Equation (4)).

$$\overline{x} = \frac{1}{M}\sum_{k=0}^{M-1} x_k \tag{4}$$

where M = Number of sample points
$\overline{x}$ = sample mean of the variable
$X_k$ = kth centered data

2. The covariance matrix is used to understand how the variables of the input data set vary from the mean for each other, something which is obtained by the equation (Equation (5)).

$$S = \frac{1}{m}\sum_{i=1}^{m}(x_i - \overline{x})(x_i - \overline{x})^T \tag{5}$$

S = covariance matrix

$\bar{x}$ = sample mean of the variable

3.   Eigenvectors and eigenvalues are the linear algebra concepts that are needed to compute from the covariance matrix to determine the principal components of the data. Eigenvalues ($\lambda$) can be obtained by the equation (Equation (6)).

$$|cov(x) - \lambda I = 0| \tag{6}$$

I = Identity matrix

4.   Eigenvectors can be determined by the equation (Equation (7)).

$$(cov(x) - \lambda I)v = 0 \tag{7}$$

V = Eigen vector

**Step-5—Interactive Supervised Classifier:** Here, stratified random sampling is used to obtain training samples for classification. Stratified random sampling is a method that involves the division of aggregation into smaller subgroups known as strata. In stratified random sampling or stratification, the strata are formed based on attributes or characteristics.

Image classification brings data classes out of a multiband raster image. It is categorized into two techniques: supervised and unsupervised. Images were classified to obtain data categories from a multiband raster image. This paper proposed a new approach, i.e., interactive supervised classification for land-use mapping. Interactive supervised classification is a tool used to reveal a supervised class without developing a signature file. It increases the speed of the categorization. Interactive supervised classification uses all the bands in the selected image with built-in pyramids. It utilizes the resolution of the actual pyramid level of the image for improved use. Building pyramids improves the display performance of raster datasets. Training sites were generated by digitizing polygons that enclosed specific land-cover features which were illustrative of desired land-cover types. The training sites from each group were then integrated into one class so that each craved cover type had one class ring from many training sites. With the training samples integrated into discrete classes, the "Interactive Supervised Classification" function was initiated to create an output image raster with cells classified and symbolized by land-cover type. Then, the entire survey field was classified into 5 land-use groups: natural vegetation, dense vegetation, urban area (without vegetation), barren land, and water. In interactive supervised classification, a pixel with the maximal probability is categorized into the proportionate class. The probability Lm is outlined as the hinder probability of a pixel associated with class k (Equation (8)).

$$Lm = P(m/S) = P(m) \times P(S/m)/\Sigma P(k) \times P(S/k) \tag{8}$$

where, P(m): the prior probability of class m. P(S/m): conditional probability to observe S from class m.

**Step-6—LULC change detection and Error matrix generation:** The error matrix, also known as the confusion matrix or transition matrix is generated, which is employed to assess remote sensing images. The diagonal elements of the matrix correspond to the correctly classified elements, upon which the classification accuracy depends. The transition matrix demonstrates several errors in the categorization procedure, allowing reinforced evaluation of maps, and increased accuracy assessment.

**Step-7—Accuracy assessment:** The classified thematic output was then examined for accuracy (Equations (9)–(16)) by calculating overall accuracy, kappa coefficient, user's, producer's accuracy, error of commission, error of omission, specificity, F-score, and false-positive rate values.

$$Overall\ Accuracy = \frac{Total\ Correctly\ Classified\ Pixels}{Total\ Number\ of\ Pixels} \tag{9}$$

$$\text{User's Accuracy} = \frac{\text{Correctly Classified Sites}}{\text{Total Number of Sites in a Row}} \tag{10}$$

$$\text{Producer's Accuracy} = \frac{\text{Correctly Classified Sites}}{\text{Total Number of Sites Classified in a Column}} \tag{11}$$

$$\text{Error of Comission} = \frac{\text{Incorrectly Classified Sites in a Row}}{\text{Total Number of Sites}} \tag{12}$$

$$\text{Error of Omission} = \frac{\text{Incorrectly Classified Sites in a Column}}{\text{Total Number of Sites}} \tag{13}$$

$$\text{Kappa Coefficient} = \frac{\text{Total Accuracy} - \text{Random Accuracy}}{1 - \text{Random Accuracy}} \tag{14}$$

$$\text{Specificity} = \frac{\text{TN}}{\text{TN} + \text{FP}} \tag{15}$$

$$\text{F} - \text{Score} = \frac{2 * \text{Precision} * \text{Recall}}{\text{Precision} + \text{Recall}} = \frac{2 * \text{TP}}{2 * \text{TP} + \text{FP} + \text{FN}} \tag{16}$$

### 3.2. Computation of LST

Land surface temperature is an essential variant of the Earth's environmental condition system. LST was assessed for 20 eras from 2000 to 2020 using thermal bands of the Landsat satellite data (ETM+, and OLI) as shown in Table 2. Three algorithms are used to calculate LST from satellite data, which are the radiative transfer model (RTM), mono-window (MW), and split-window (SW) algorithms. RTM technique produces a top-quality outcome [51]. However, the required dedication of the radiosonde interest of atmospheric elements throughout the passage of the satellite makes the set of rules much less endorsed for SW and MW. However, SW requires two thermal bands, which produces some unpredictability. Thus, the MW algorithmic rule is preferred based on the thermal radius [52,53]. Therefore, in this research, LST was enumerated by exploitation of the MW algorithm formulated by Qin et al. (2001). Figure 5 represents the flow chart for the computation of LST. Thermal conversion constants of ETM+ and OLI sensors are shown in Table 2, where K1 and K2 are thermal conversion constants of the TIRS bands and $Q_{cal}$–quantized and calibrated standard product pixel value (DN).

**Table 2.** Thermal conversion constants.

| Constants | ETM+ | OLI |
|---|---|---|
| $K_1$ | 666.09 | 774.8853 |
| $K_2$ | 1282.71 | 1321.0789 |
| $L_{max}$ | 12.65 | 22.00180 |
| $L_{min}$ | 3.200 | 0.10033 |
| $Q_{cal\,max}$ | 255 | 65,535 |
| $Q_{cal\,min}$ | 1 | 1 |

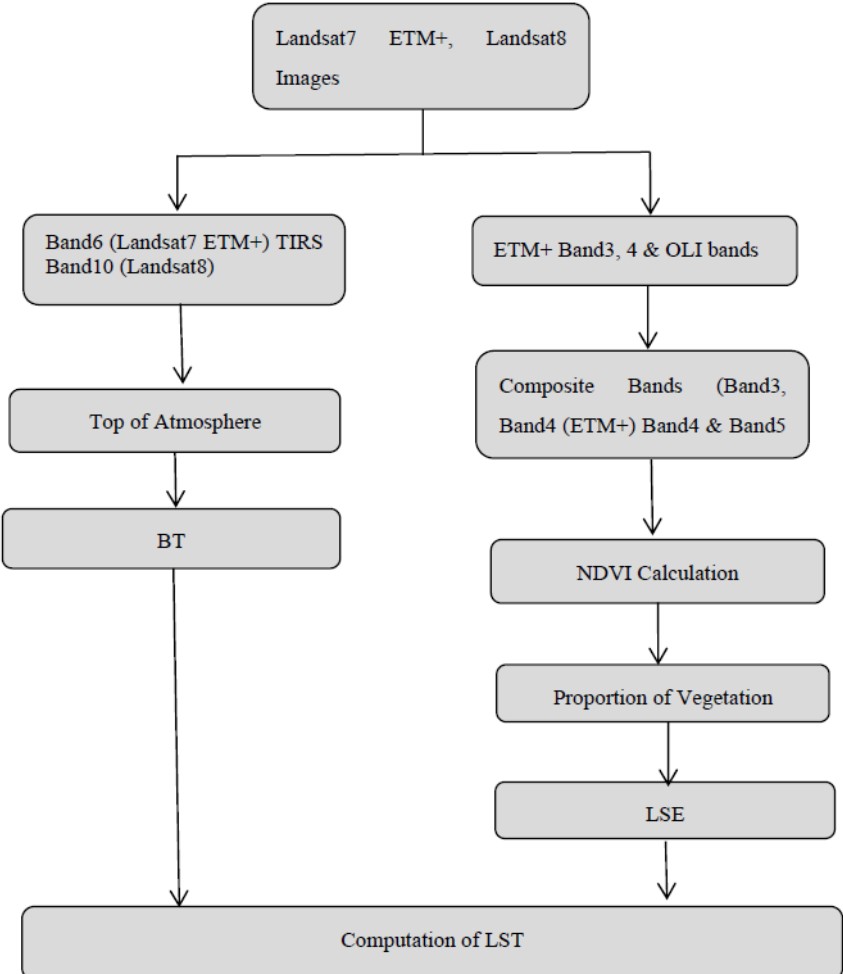

**Figure 5.** Flowchart for Computation of LST.

The step-by-step process for LST calculation is given below:

**Step-1:** Radiance is the "flux of energy per solid angle leaving a unit surface area in a given direction." The initial step is to transform the digital number (DN), i.e., the number of pixels in an image, into top of atmosphere (ToA) radiance. For Landsat images, spectral radiance at the sensor's aperture is given by the equation (Equation (17)).

$$L_\lambda = M_L * Q_{cal} + A_L + O_i \tag{17}$$

where

$L_\lambda$—ToA spectral radiance

$M_L$—Band-specific multiplicative rescaling factor from the metadata (Radiance_mult_Band_X, where X is the band number 10)

$A_L$—Band-specific additive rescaling factor from the metadata (Radiance_add_Band_10)

$Q_{cal}$—Quantized and calibrated standard product pixel value (DN)

$O_i$—Correction value of Band 10, which is 0.29

**Step-2:** Brightness temperature (BT) is a measure of the power of microwave radiation traveling upwards from the upper atmosphere to the satellite, expressed in units of equivalent blackbody temperature. Spectral radiance data conceivably changed into the top of atmosphere brightness temperature using invariant thermal values in the metadata file. The brightness temperature transformation is given in the equation (Equation (1)).

$$BT = \left[ \frac{K_2}{\ln(K_1 + L_\lambda + 1)} \right] - 273.15 \tag{18}$$

where

$L_\lambda$—TOA spectral radiance

$K_1$, $K_2$—Band-specific thermal conversion from metadata (K1-Constant_Band_10, K2-Constant_Band_10)

273.15 = Convert Kelvin to $0^0$ Celsius.

**Step-3:** NDVI is the normalized difference vegetation index and may be measured with the aid of using the equation (Equation (19))

$$NDVI = \frac{NIR - RED}{NIR + RED} \tag{19}$$

**Step-4**: Proportion of vegetation $P_V$ is outlined as the ratio of the vertical projection region of vegetation (including leaves, root word, and subdivision) on the ground to the whole vegetation region and calculated as the equation (Equation (20))

$$P_v = \left[ \frac{NDVI - NDVI_{Min}}{NDVI_{Max} - NDVI_{Min}} \right]^2 \tag{20}$$

From the above calculated NDVI, we can obtain the NDVImin and NDVImax, used further to calculate the proportion of vegetation ($P_V$).

**Step-5:** The next step is to calculate land surface emissivity (LSE). Land surface emissivity (LSE), as an inherent property of natural substantial, is frequently used as an index of material composition and is calculated as the equation (Equation (21)).

$$E = 0.004 * P_v + 0.986 \tag{21}$$

where E is Emissivity

Pv is the proportion of vegetation that is premeditated using the NDVI value

**Step-6:** The last step is to calculate LST. By using the above-calculated values in the LST equation, we can obtain the LST of the required region. The formula to calculate LST (Equation (22)) is as follows:

$$LST = \frac{B_T}{(1 + (\lambda * B_T/C_2) \ln(E))} \tag{22}$$

where $\lambda$ is the emitted radiance wavelength of 10.8 µm

$C_2$ = 14,388 µmK

E is the land surface Emissivity

## 4. Results

### 4.1. LULC Analysis

LULC of the survey region is classified as built-up (settlements, business enterprises, and some other substructure), dense vegetation (forests), vegetation (agriculture, and woody plant), water body, barren, open land, and rocks. Figures 6–8 represents the classified images of the three regions of 2000, 2005, 2010, 2015, and 2020. LULC consequences of the survey region are represented in Tables 3–5. The error matrix encapsulates that the built-up area has magnified from 2950.24 hectares in 2000 to 9503.3 hectares in 2020 (3259.73 Ha in 2005, 4288.4 Ha in 2010, and 6122.5 Ha in 2015). The total built-up area increased throughout the study period, exhibiting an exponent increase. Dense vegetation decreased from 25,517.5 ha in 2000 to 13,950.27 ha in 2020. Vegetation showed an increasing trend from 5908.07 Ha in 2000 to 11,898.7 Ha in 2020. In the Tirupati region, the transition matrix summarizes that the dense vegetation, vegetation, and built-up area have magnified to 1.8%, 7.79%, and 8.54% from 2000 to 2020, and that water and barren land have decreased

by −5.56% and −12.2%. In the Visakhapatnam region, the dense vegetation, vegetation, and built-up area have magnified to 0.8%, 0.08%, and 32.51% from 2000 to 2020, and water and barren land decreased by −23.42% and −9.98%. In Vijayawada region, the vegetation, water and built-up area have increased to 16.04%, 1.8% and 17.55% from 2000 to 2020 and the dense vegetation, water, barren land have decreased by −30.98, −4.5%.

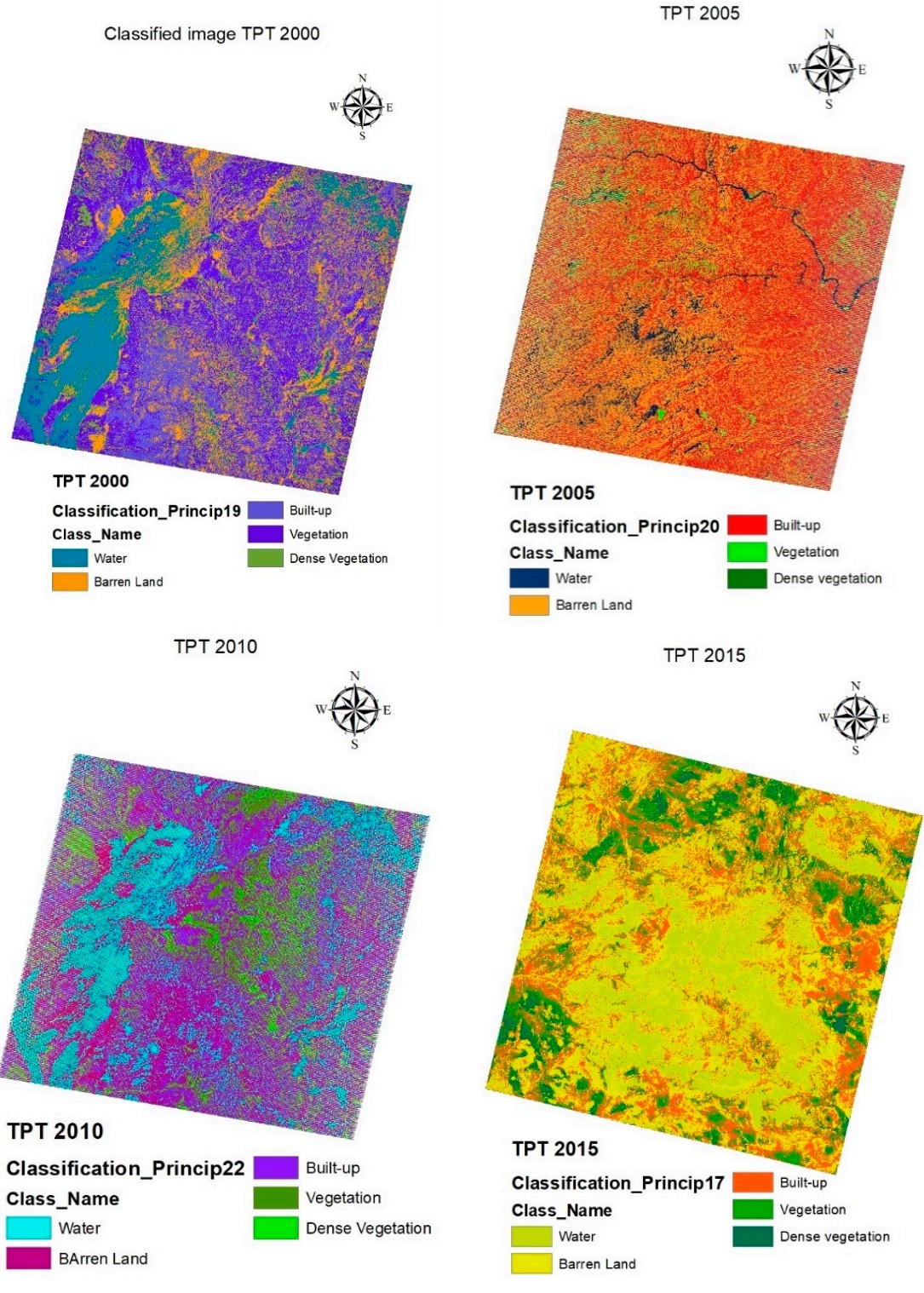

**Figure 6.** *Cont.*

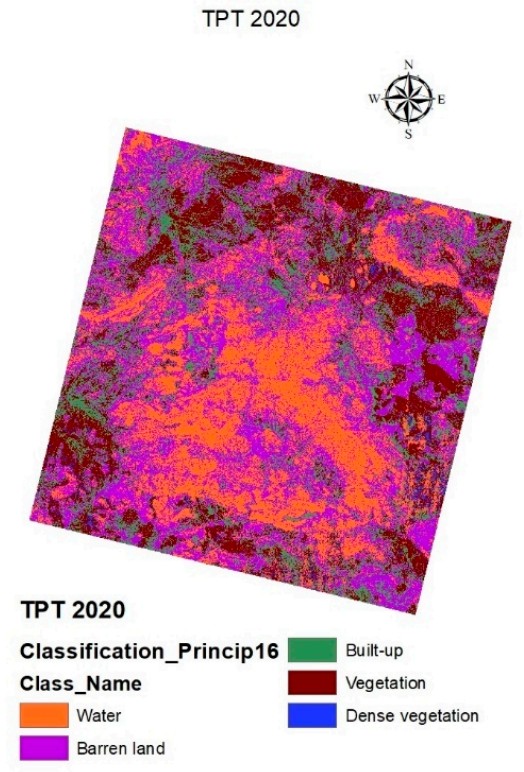

**Figure 6.** Classified images of Tirupati area of the years 2000, 2005, 2010, 2015 and 2020.

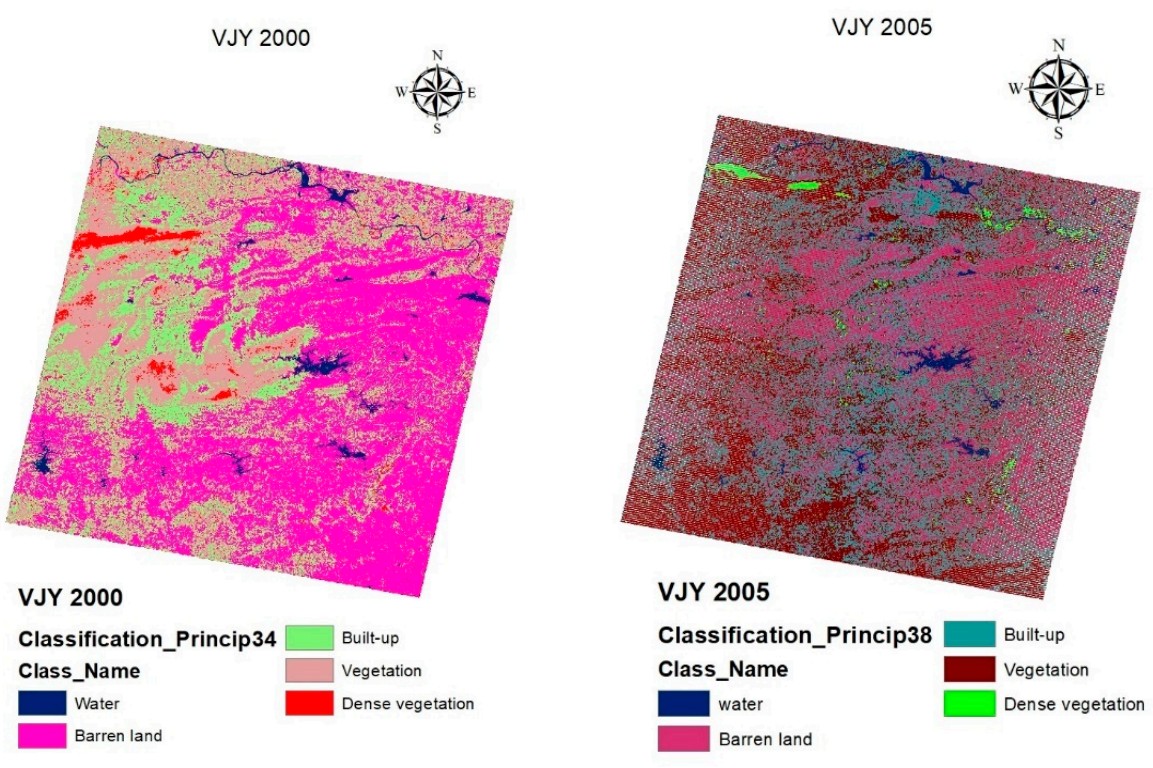

**Figure 7.** *Cont.*

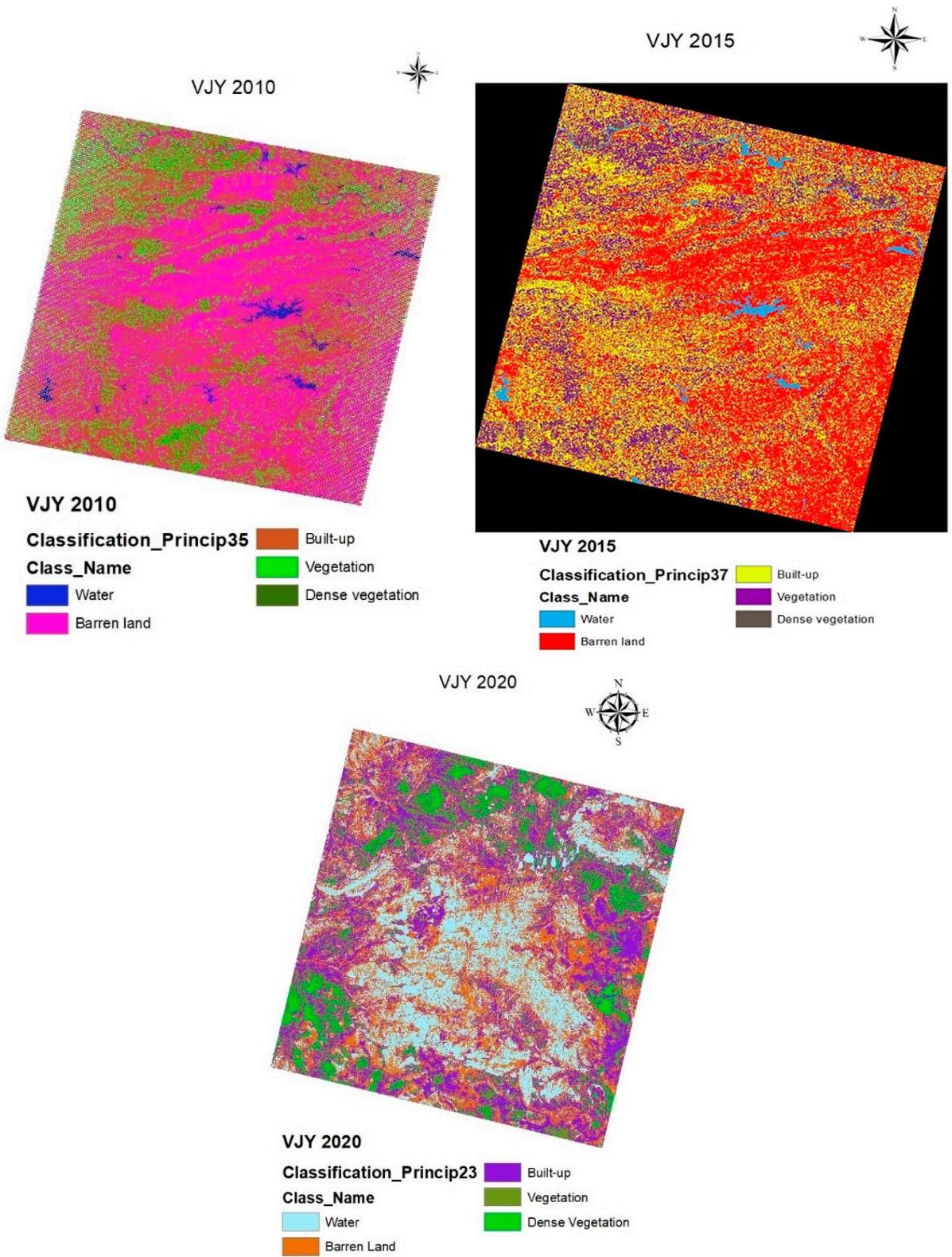

**Figure 7.** Classified images of Vijayawada area of the years 2000, 2005, 2010, 2015 and 2020.

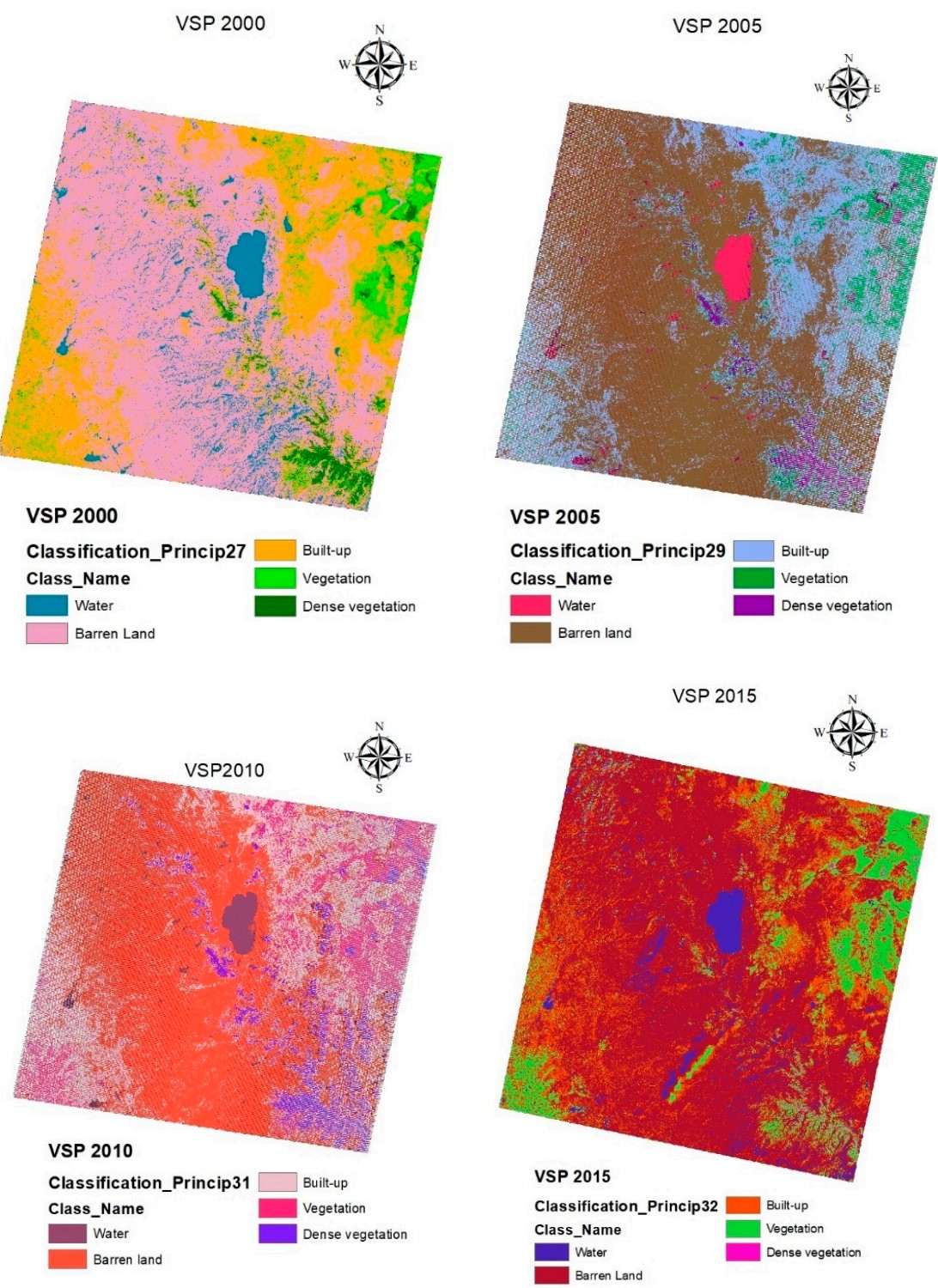

**Figure 8.** *Cont.*

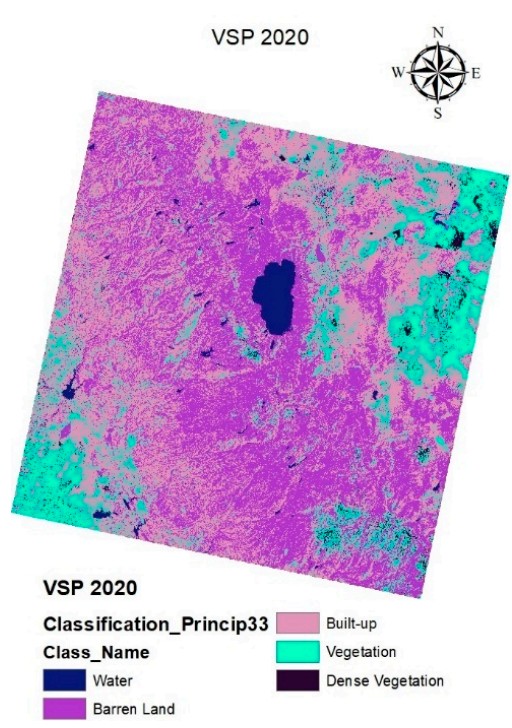

**Figure 8.** Classified images of Visakhapatnam area of the years 2000, 2005, 2010, 2015 and 2020.

**Table 3.** LULC consequences of the study area (Tirupati).

| LULC Category | Area (Hectares) | | | | | | | | | | Change in Area (Hectares) | Change in Area (%) |
|---|---|---|---|---|---|---|---|---|---|---|---|---|
| | Year (2000) | Area (%) | Year (2005) | Area (%) | Year (2010) | Area (%) | Year (2015) | Area (%) | Year (2020) | Area (%) | | |
| Dense vegetation | 5857.62 | 16.54 | 5482.53 | 15.49 | 5828.29 | 16.47 | 6467.46 | 18.27 | 6490.25 | 18.34 | 632.63 | 1.80 |
| Vegetation | 9268.07 | 26.18 | 9419.84 | 26.61 | 10,300.05 | 29.10 | 11,090.49 | 31.33 | 12,023.85 | 33.97 | 2755.78 | 7.79 |
| Urban | 11,074.08 | 31.28 | 11,637.09 | 32.88 | 11,981.78 | 33.85 | 12,129.59 | 34.27 | 14,094.56 | 39.82 | 3020.48 | 8.54 |
| Barren Land | 6629.97 | 18.73 | 6491.48 | 18.34 | 5380.86 | 15.20 | 4206.45 | 11.88 | 2205.70 | 6.23 | −4424.27 | −12.50 |
| Water | 2564.62 | 7.2 | 2363.42 | 6.68 | 1903.38 | 5.38 | 1500.38 | 4.24 | 579.99 | 1.64 | −1984.63 | −5.56 |
| Total | 35,394.38 | | 35,394.38 | | 35,394.38 | | 35,394.38 | | 35,394.38 | | | |

**Table 4.** LULC consequences of the study area (Visakhapatnam).

| LULC Category | Area (Hectares) | | | | | | | | | | Change in Area (Hectares) | Change in Area (%) |
|---|---|---|---|---|---|---|---|---|---|---|---|---|
| | Year (2000) | Area (%) | Year (2005) | Area (%) | Year (2010) | Area (%) | Year (2015) | Area (%) | Year (2020) | Area (%) | | |
| Dense vegetation | 688.04 | 2.88 | 630.83 | 2.64 | 973.39 | 4.07 | 925.48 | 3.87 | 880.28 | 3.68 | 192.24 | 0.80 |
| Vegetation | 5018.57 | 20.99 | 4910.27 | 20.53 | 4750.29 | 19.86 | 4040.93 | 16.90 | 5037.82 | 21.07 | 19.25 | 0.08 |
| Urban | 8425.71 | 35.23 | 11,501.08 | 48.09 | 12,997.44 | 54.35 | 14,221.88 | 59.47 | 16,200.84 | 67.75 | 7775.67 | 32.51 |
| Barren Land | 2888.00 | 12.08 | 2240.47 | 9.37 | 2198.72 | 9.19 | 2140.62 | 8.95 | 501.67 | 2.10 | −2386.33 | −9.98 |
| Water | 6893.29 | 28.83 | 4630.96 | 19.37 | 2993.76 | 12.52 | 2584.70 | 10.81 | 1292.99 | 5.41 | −5600.27 | −23.42 |
| Total | 23,913.62 | | 23,913.62 | | 23,913.62 | | 23,913.62 | | 23,913.62 | | | |

**Table 5.** LULC consequences of the study area (Vijayawada).

| LULC Category | Area (Hectares) | | | | | | | | | | Change in Area (Hectares) | Change in Area (%) |
|---|---|---|---|---|---|---|---|---|---|---|---|---|
| | Year (2000) | Area (%) | Year (2005) | Area (%) | Year (2010) | Area (%) | Year (2015) | Area (%) | Year (2020) | Area (%) | | |
| Dense vegetation | 25,517.5 | 68.34 | 22,889 | 61.30 | 21,013.5 | 56.28 | 18,801.5 | 50.35 | 13,950.27 | 37.36 | −11,567.23 | −30.98 |
| Vegetation | 5908.07 | 15.82 | 8350.53 | 22.36 | 9136.9 | 24.47 | 10,045.1 | 26.90 | 11,898.7 | 31.87 | 5990.63 | 16.04 |
| Urban | 2950.24 | 7.90 | 3259.73 | 8.73 | 4288.4 | 11.48 | 6122.5 | 16.40 | 9503.3 | 25.45 | 6553.06 | 17.55 |
| Barren Land | 2754.47 | 7.38 | 2332.37 | 6.25 | 2123.17 | 5.69 | 1515.27 | 4.06 | 1075.1 | 2.88 | −1679.37 | −4.50 |
| Water | 209.09 | 0.56 | 507.74 | 1.36 | 777.4 | 2.08 | 855 | 2.29 | 912 | 2.44 | 702.91 | 1.88 |
| Total | 37,339.37 | | 37,339.37 | | 37,339.37 | | 37,339.37 | | 37,339.37 | | | |

The error matrix or transition matrix generated is employed to assess remote sensing images. The diagonal elements of the matrix correspond to the correctly classified elements upon which the classification accuracy depends. The error matrix or transition matrix demonstrates several errors in the classification procedure, allowing reinforced evaluation of maps, and increased accuracy assessment. Kappa quality appraisal is accomplished to show the divergence between the existent statement and the declaration anticipated via the means of risk and is a vital empirical approach in reading remote sensing quantifiable data. The sample of an error matrix is represented in Table 6. In remote sensing, ground truth refers to data gathered on location. Ground truth allows image data to be accompanied by real features and materials on the ground. In the case of a classified image, it allows supervised classification to assist in finding out the accuracy of the classification performed by the remote sensing software and thus decrease errors in the classification, such as errors of commission and errors of omission. Here, in this study, field visits to the study areas were undertaken, during which some ground truths were collected specially for undeveloped areas and the physical objects of classes were recorded by GPS. According to these two sources, i.e., field visits and GPS, different ground truths of land-cover classification were recorded manually. A good rule of thumb is to collect a minimum of 50 samples for each land-cover category. If the area is peculiarly large or the classification system incorporates more categories, then the minimum number of samples should be 75 to 100 per class [54]. Based on the rule of thumb, 90 to 150 samples are considered for each land-cover category of the area of interest (AoI).

The overall accuracy and kappa coefficients of the three regions for the years 2000, 2005, 2010, 2015, and 2020 are represented in Table 7. Based on the transition matrix, the overall classification quality of the Vijayawada region was ascertained as 96.4%, 94.5%, 93%, 91%, 88%, with individual kappa coefficients of 0.95, 0.93, 0.91, 0.89, and 0.87 approximately for the years 2020, 2015, 2010, 2005 and 2000, respectively. The estimated kappa coefficient shows the excellent and time-tested output from the classifier for all the years. Based on the error matrix, the overall classification accuracy of the Visakhapatnam region was ascertained as 97%, 94.5%, 93%, 92.5%, and 89.3%, with their individual kappa coefficients of 0.96, 0.92, 0.91, 0.9, and 0.87 approximately for the years 2020, 2015, 2010, 2005 and 2000, respectively. Overall accuracy represents what equipoise of the web sites is mapped correctly. Based on the error matrix generated, the overall classification accuracy of Tirupati region was ascertained as 96.8%, 94.5%, 92.5%, 91%, and 90.4% with their individual kappa coefficients as 0.96, 0.92, 0.91, 0.9, and 0.89, approximately, for the years 2020, 2015, 2010, 2005 and 2000. Maximum urban development occurred between 2000 to 2020, and LULC changes are graphically represented in Figure 9. This improvement inside the built-up area is assigned to the brand-new industrial business enterprise and academic governance in and across the towns with a lack of flora and a few different classes.

**Table 6.** Transition matrix of Vijayawada for the time-period 2020.

| | Class ID | Reference Data | | | | | Ground Truth Points | User's Accuracy (%) | Error of Commission | Specificity | F-Score (%) | False-Positive Rate |
|---|---|---|---|---|---|---|---|---|---|---|---|---|
| | | 1 | 2 | 3 | 4 | 5 | | | | | | |
| Classified Data | 1 | 98 | 3 | 2 | 0 | 0 | 103 | 95.1 | 4.8 | 0.98 | 95.5 | 0.02 |
| | 2 | 4 | 96 | 2 | 0 | 0 | 102 | 94.1 | 5.8 | 0.98 | 94.1 | 0.02 |
| | 3 | 0 | 3 | 90 | 0 | 0 | 93 | 96.7 | 3.2 | 0.98 | 94.6 | 0.02 |
| | 4 | 0 | 0 | 3 | 91 | 0 | 94 | 96.8 | 3.1 | 1 | 98.3 | 0 |
| | 5 | 0 | 0 | 0 | 0 | 89 | 89 | 100 | 0 | 1 | 100 | 0 |
| Ground truth points | | 102 | 102 | 97 | 91 | 89 | 481 | | | | | |
| Producer's Accuracy (%) | | 96 | 94.1 | 92.7 | 100 | 100 | | | | | | |
| Error of Omission | | 3.9 | 5.8 | 7.2 | 0 | 0 | | | | | | |

1—Dense Vegetation, 2—Vegetation, 3—Built-up, 4—Barren Land, 5—Water. Overall Accuracy = 96.4%; Kappa Coefficient = 0.95.

**Table 7.** Overall Accuracy and Kappa Coefficient for the period 2000, 2005, 2010, 2015, and 2020 of Region of Interest.

| Region of Interest | 2000 | | 2005 | | 2010 | | 2015 | | 2020 | |
|---|---|---|---|---|---|---|---|---|---|---|
| | Overall Accuracy (%) | Kappa Coefficient | Overall Accuracy (%) | Kappa Coefficient | Overall Accuracy (%) | Kappa Coefficient | Overall Accuracy (%) | Kappa Coefficient | Overall Accuracy (%) | Kappa Coefficient |
| Vijayawada (VJY) | 88 | 0.86 | 91.5 | 0.89 | 93 | 0.91 | 94.5 | 0.93 | 96.4 | 0.95 |
| Visakhapatnam (VSP) | 89.3 | 0.87 | 92.5 | 0.9 | 93 | 0.91 | 94.5 | 0.92 | 97 | 0.96 |
| Tirupati (TPT) | 90.4 | 0.89 | 91 | 0.9 | 92.5 | 0.91 | 94.5 | 0.92 | 96.8 | 0.96 |

The producer's accuracy tells how real attributes are efficaciously proven on labeled maps based on the mapmaker's viewpoint (i.e., the producer). The user's accuracy tells how the beauty on the map may be reached on the ground based on the user's path; this is based on the dependability of the map. The producer's accuracy for each category was higher than 80%. The user's accuracy for five categories was more than 85%. The errors generated (i.e., error of commission and error of omission) are less than 20%. The false-positive rate is a measure of how many results get predicted as positive out of all the negative cases. F-score is a way to measure classification accuracy based on recall and precision. The higher an F-score, the more accurate the classification is. F-score obtained for five categories was more than 80% which represents significant classification. Specificity is the ability of a test to correctly identify classes. The user's accuracy, producer's accuracy, error of commission, error of omission, specificity, F-score, and false-positive rate are represented in Tables 8–14.

The Comparison of the proposed method with other techniques is shown in Table 15.

As can be seen in Table 15, it is concluded that the proposed method shows the highest overall accuracy and kappa coefficient in comparison to the earlier described methods. It works efficiently on all types of satellite images. The proposed method is better than the earlier methods, as it takes the maximum amount of assessment parameters for the study under the circumstance.

In the present study, RED, near-infrared, and thermal infrared images from the TM Landsat 7 and OLI/TIRS-Landsat 8 sensors were utilized for LULC classification and to calculate NDVI and LST. MW algorithm was used to retrieve the LST of images accumulated by the sensors throughout the series (2000–2020), and LULC classification was accomplished for the years 2000, 2005, 2010, 2015, and 2020 to verify the consequence of LULC classes on the temperature from NDVI on a temporal and spatial scale. This paper presents the spatially continuous regional LULC description of the regional data of Andhra Pradesh state, India. The maps also cover, with an annual temporal resolution, a period

of 20 years with aggravated LULC transformations. Additionally, this work provides a methodological alternative for the continuous description of LULC, comparatively fast and with low cost, over large areas, and with advanced periodicity. The overall accuracy of the maps and the low and similar values of the omission and commission errors indicate a low level of sub and over-appraisal of the different class coverage. The PCA-based feature extraction approach along with NDVI, Interactive Supervised Classifier has performed better (OA: 97%) compared to other conceptualizations.

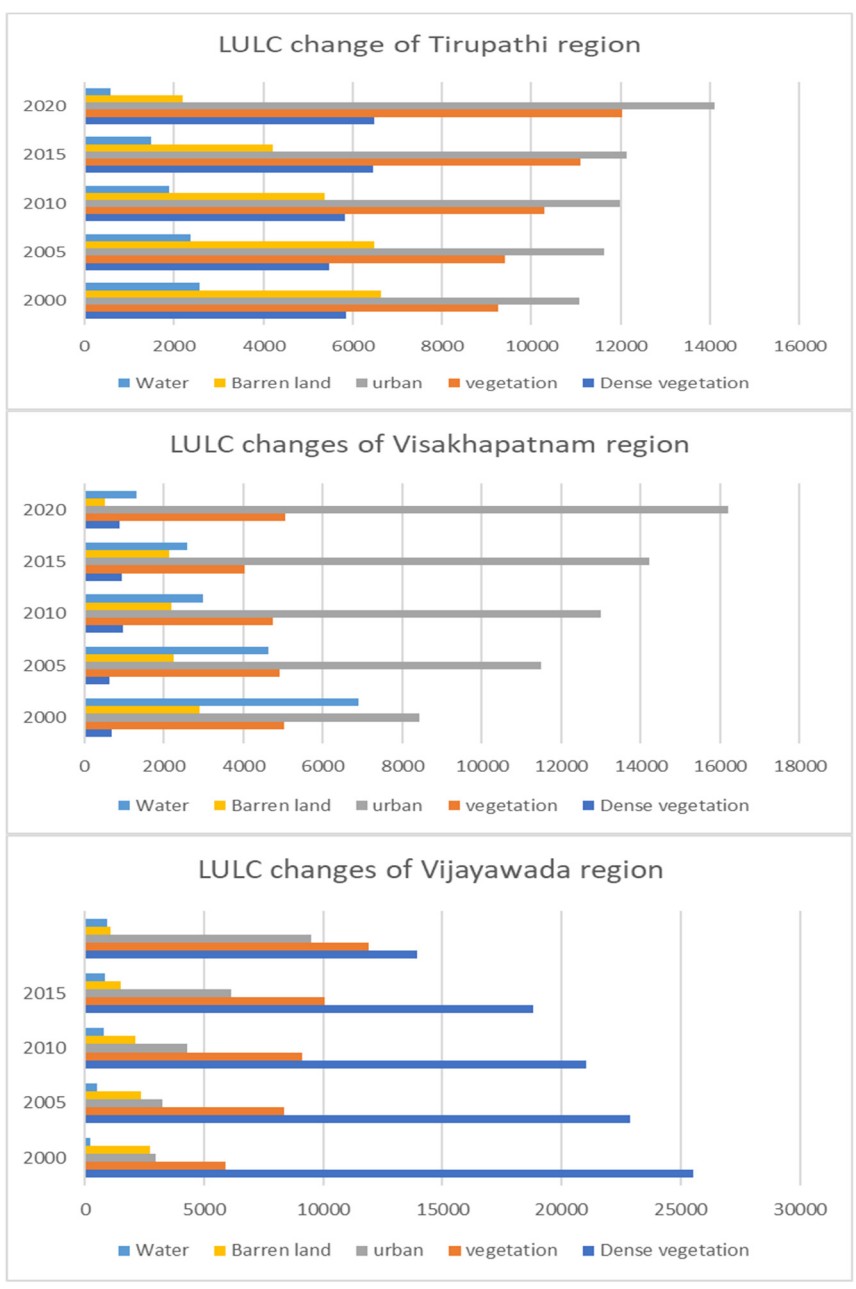

**Figure 9.** Graphical representation of LULC changes in Tirupati, Visakhapatnam, and Vijayawada regions.

**Table 8.** User's Accuracy (%) for the period 2000, 2005, 2010, 2015, and 2020 of Region of Interest.

| Type of Land Cover | 2000 | | | 2005 | | | 2010 | | | 2015 | | | 2020 | | |
|---|---|---|---|---|---|---|---|---|---|---|---|---|---|---|---|
| | VJY | VSP | TPT | VJY | VSP | TPT | VJY | VSP | TPT | VJY | VSP | TPT | VJY | VSP | TPT |
| 1 | 87.5 | 86.8 | 89.1 | 91.8 | 91.2 | 90 | 92.5 | 92.5 | 91.2 | 93.8 | 93.8 | 93.8 | 95.1 | 97.6 | 99 |
| 2 | 89.4 | 88.6 | 89.4 | 92.7 | 89.6 | 93.8 | 91 | 91 | 89.6 | 94.4 | 93 | 93 | 94.1 | 96.4 | 95.2 |
| 3 | 85.3 | 88.7 | 88.7 | 88.8 | 92 | 90.9 | 92 | 92 | 92 | 91.6 | 94.8 | 94.8 | 96.7 | 98.8 | 97 |
| 4 | 87.5 | 91.6 | 91.6 | 92.7 | 95.8 | 90.7 | 95.8 | 95.8 | 95.8 | 96.2 | 94.8 | 94.8 | 96.8 | 96.4 | 97 |
| 5 | 91.8 | 91 | 93.4 | 91.3 | 93.9 | 91.3 | 94 | 94 | 93.9 | 97 | 95.9 | 95.9 | 100 | 97.6 | 97.8 |

1—Dense Vegetation; 2—Vegetation; 3-Built-up; 4—Barren Land; 5—Water.

**Table 9.** Producer's Accuracy (%) for the period 2000, 2005, 2010, 2015, and 2020 of Region of Interest.

| Type of Land Cover | 2000 | | | 2005 | | | 2010 | | | 2015 | | | 2020 | | |
|---|---|---|---|---|---|---|---|---|---|---|---|---|---|---|---|
| | VJY | VSP | TPT | VJY | VSP | TPT | VJY | VSP | TPT | VJY | VSP | TPT | VJY | VSP | TPT |
| 1 | 92.5 | 92.7 | 92.7 | 96.7 | 93.1 | 96.7 | 93.1 | 93.1 | 93.1 | 95.8 | 97.8 | 97.8 | 96 | 97.6 | 98 |
| 2 | 80.8 | 83.3 | 83.3 | 85.7 | 88.4 | 87.6 | 88.5 | 88.5 | 88.4 | 89.4 | 93.9 | 93.9 | 94.1 | 97.6 | 97 |
| 3 | 82.6 | 85.9 | 85.9 | 88 | 90.1 | 85.7 | 92.6 | 92.6 | 90.1 | 94.9 | 88.5 | 88.5 | 92.7 | 97.6 | 95 |
| 4 | 92.9 | 93.2 | 93.2 | 90.8 | 91.5 | 90.7 | 91.5 | 91.5 | 91.5 | 93.5 | 94.8 | 94.8 | 100 | 96.4 | 98 |
| 5 | 94.1 | 98.2 | 98.2 | 97.7 | 100 | 97.7 | 100 | 100 | 100 | 100 | 98 | 98 | 100 | 97.6 | 96 |

1—Dense Vegetation; 2—Vegetation; 3-Built-up; 4-Barren Land; 5—Water.

**Table 10.** Error of Commission for the period 2000, 2005, 2010, 2015, and 2020 of Region of Interest.

| Type of Land Cover | 2000 | | | 2005 | | | 2010 | | | 2015 | | | 2020 | | |
|---|---|---|---|---|---|---|---|---|---|---|---|---|---|---|---|
| | VJY | VSP | TPT | VJY | VSP | TPT | VJY | VSP | TPT | VJY | VSP | TPT | VJY | VSP | TPT |
| 1 | 12.5 | 13.1 | 10.85 | 8.1 | 8.7 | 10 | 7.4 | 7.4 | 8.7 | 6.1 | 6.1 | 6.1 | 4.8 | 2.3 | 1 |
| 2 | 10.5 | 11.3 | 10.56 | 7.2 | 10.3 | 6.1 | 8.9 | 8.9 | 10.3 | 5.5 | 7 | 7 | 5.8 | 3.5 | 6.5 |
| 3 | 14.6 | 11.2 | 11.29 | 11.1 | 8 | 9 | 8 | 8 | 8 | 8.3 | 5.1 | 5.1 | 3.2 | 1.1 | 3 |
| 4 | 12.5 | 8.3 | 8.33 | 7.2 | 4.1 | 9.2 | 4.1 | 4.1 | 4.1 | 3.7 | 5.1 | 5.1 | 3.1 | 3.5 | 3 |
| 5 | 8.13 | 8.9 | 6.5 | 8.6 | 6 | 8.6 | 6 | 6 | 6 | 3 | 4 | 4 | 0 | 2.3 | 2.1 |

1—Dense Vegetation; 2—Vegetation; 3—Built-up; 4—Barren Land; 5—Water.

**Table 11.** Error of Omission for the period 2000, 2005, 2010, 2015, and 2020 of Region of Interest.

| Type of Land Cover | 2000 | | | 2005 | | | 2010 | | | 2015 | | | 2020 | | |
|---|---|---|---|---|---|---|---|---|---|---|---|---|---|---|---|
| | VJY | VSP | TPT | VJY | VSP | TPT | VJY | VSP | TPT | VJY | VSP | TPT | VJY | VSP | TPT |
| 1 | 7.4 | 7.2 | 7.2 | 3.2 | 6.8 | 3.2 | 6.8 | 6.8 | 6.8 | 4.1 | 2.1 | 2.1 | 3.9 | 2.3 | 2 |
| 2 | 19.1 | 16.6 | 16.6 | 14.2 | 11.5 | 12.3 | 11.4 | 11.4 | 11.5 | 4.6 | 6.1 | 6.1 | 5.8 | 2.3 | 3 |
| 3 | 17.3 | 14 | 14 | 12 | 9.8 | 14.2 | 7.3 | 7.3 | 9.8 | 5.1 | 11.4 | 11.4 | 7.2 | 2.3 | 5 |
| 4 | 7.0 | 6.7 | 6.7 | 9.1 | 8.4 | 9.2 | 8.4 | 8.4 | 8.4 | 6.4 | 5.1 | 5.1 | 0 | 1.1 | 2 |
| 5 | 5.8 | 1.7 | 1.7 | 2.2 | 0 | 2.2 | 0 | 0 | 0 | 0 | 2 | 2 | 0 | 2.3 | 4 |

1—Dense Vegetation; 2—Vegetation; 3—Built-up; 4—Barren Land; 5—Water.

**Table 12.** Specificity for the period 2000, 2005, 2010, 2015, and 2020 of Region of Interest.

| Type of Land Cover | 2000 | | | 2005 | | | 2010 | | | 2015 | | | 2020 | | |
|---|---|---|---|---|---|---|---|---|---|---|---|---|---|---|---|
| | VJY | VSP | TPT | VJY | VSP | TPT | VJY | VSP | TPT | VJY | VSP | TPT | VJY | VSP | TPT |
| 1 | 0.97 | 0.98 | 0.98 | 0.99 | 0.98 | 0.97 | 0.98 | 0.98 | 0.98 | 0.98 | 0.99 | 0.99 | 0.98 | 0.99 | 1 |
| 2 | 0.94 | 0.95 | 0.95 | 0.95 | 0.97 | 0.98 | 0.97 | 0.97 | 0.97 | 0.97 | 0.98 | 0.98 | 0.98 | 0.99 | 1 |
| 3 | 0.95 | 0.94 | 0.96 | 0.96 | 0.97 | 0.97 | 0.98 | 0.98 | 0.97 | 0.98 | 0.97 | 0.97 | 0.98 | 0.99 | 0.98 |
| 4 | 0.98 | 0.98 | 0.98 | 0.97 | 0.97 | 0.97 | 0.97 | 0.97 | 0.97 | 0.98 | 0.98 | 0.98 | 1 | 0.99 | 0.98 |
| 5 | 0.98 | 0.99 | 0.99 | 0.99 | 1 | 0.98 | 1 | 1 | 1 | 1 | 0.99 | 0.99 | 1 | 0.99 | 0.99 |

1—Dense Vegetation; 2—Vegetation; 3—Built-up; 4—Barren Land; 5—Water

**Table 13.** F-Score (%) for the period 2000, 2005, 2010, 2015, and 2020 of Region of Interest.

| Type of Land Cover | 2000 | | | 2005 | | | 2010 | | | 2015 | | | 2020 | | |
|---|---|---|---|---|---|---|---|---|---|---|---|---|---|---|---|
| | VJY | VSP | TPT | VJY | VSP | TPT | VJY | VSP | TPT | VJY | VSP | TPT | VJY | VSP | TPT |
| 1 | 89.9 | 90.8 | 90.8 | 94.1 | 92.1 | 93.2 | 92.7 | 92.7 | 92.1 | 94.7 | 95.7 | 95.7 | 95.5 | 97.6 | 98.4 |
| 2 | 84.8 | 86.2 | 86.2 | 89.1 | 88.9 | 90.5 | 89.7 | 89.7 | 88.9 | 91.8 | 93.4 | 93.4 | 94.1 | 97 | 95.8 |
| 3 | 83.9 | 87.2 | 87.2 | 88.39 | 91.0 | 88.2 | 92.2 | 92.2 | 91.0 | 93.2 | 91.5 | 91.5 | 94.6 | 97.6 | 96 |
| 4 | 90.1 | 92.4 | 92.4 | 91.7 | 93.6 | 90.7 | 93.6 | 93.6 | 93.6 | 94.8 | 94.8 | 94.8 | 98.3 | 96.5 | 97.4 |
| 5 | 92.3 | 95.7 | 95.7 | 94.3 | 96.8 | 94.3 | 96.9 | 96.9 | 96.8 | 98.4 | 96.9 | 96.9 | 100 | 97.6 | 96.8 |

1—Dense Vegetation; 2—Vegetation; 3—Built-up; 4—Barren Land; 5—Water

**Table 14.** False-Positive Rate for the period 2000, 2005, 2010, 2015, and 2020 of Region of Interest.

| Type of Land Cover | 2000 | | | 2005 | | | 2010 | | | 2015 | | | 2020 | | |
|---|---|---|---|---|---|---|---|---|---|---|---|---|---|---|---|
| | VJY | VSP | TPT | VJY | VSP | TPT | VJY | VSP | TPT | VJY | VSP | TPT | VJY | VSP | TPT |
| 1 | 0.03 | 0.02 | 0.02 | 0.01 | 0.02 | 0.03 | 0.02 | 0.02 | 0.02 | 0.02 | 0.01 | 0.01 | 0.02 | 0.01 | 0 |
| 2 | 0.06 | 0.05 | 0.05 | 0.05 | 0.03 | 0.02 | 0.03 | 0.03 | 0.03 | 0.03 | 0.02 | 0.02 | 0.02 | 0.01 | 0 |
| 3 | 0.05 | 0.06 | 0.04 | 0.04 | 0.03 | 0.03 | 0.02 | 0.02 | 0.03 | 0.02 | 0.03 | 0.03 | 0.02 | 0.01 | 0.02 |
| 4 | 0.02 | 0.02 | 0.02 | 0.03 | 0.03 | 0.03 | 0.03 | 0.03 | 0.03 | 0.02 | 0.02 | 0.02 | 0 | 0.01 | 0.02 |
| 5 | 0.02 | 0.01 | 0.01 | 0.01 | 0 | 0.02 | 0 | 0 | 0 | 0 | 0.01 | 0.01 | 0 | 0.01 | 0.01 |

1—Dense Vegetation; 2—Vegetation; 3—Built-up; 4—Barren Land; 5—Water.

**Table 15.** Comparison of the proposed method with other techniques.

| Author (Year) | Method | Type of Images Used | Evaluation Parameters | |
|---|---|---|---|---|
| | | | Accuracy (%) | Kappa Coefficient |
| Sundarakumar et al. [36] (2012) | Obtained LULC changes and urban sprawl research of Vijayawada city of years 1990 and 2009 ML Classifier | Landsat ETM+ | 86.67 (1990) 85 (2009) | 0.8 (1990) 0.78 (2009) |
| K. Sundara et al. [39] (2012) | Estimated Land Surface Temperature of Landsat ETM+ images of year 2001using Mono Window Algorithm Obtained LULC changes with the help of ML Classifier | Landsat ETM+ | 80 | 0.729 |

**Table 15.** *Cont.*

| Author (Year) | Method | Type of Images Used | Evaluation Parameters | |
|---|---|---|---|---|
| | | | Accuracy (%) | Kappa Coefficient |
| Kiran Yerrakula et al. [40] (2014) | Analyzed urban sprawl changes and detected LULC changes in Vijayawada city using Minimum Distance Classifier | Landsat8 | 67.19 | 0.6405 |
| Vani, M. et al. [3] (2018) | Assessed spatio-temporal modifications in LULC, urban sprawl, and LST in the vicinity of Vijayawada city in the years 1990, 2000, 2010, and 2018 using NDVI, ML Classifier | Landsat ETM+, Landsat8 | 94.33 (2018), 93.07 (2010), 92.0 (2000), and 87.0 (1990) | 0.94 (2018), 0.87 (2010), 0.88 (2000), and 0.81 (1990) |
| GN Vivekananda et al. [34] (2020) | Accuracy assessment of the Tirupati region was performed in 1978 and 2018 with the help of ML Classifier | Landsat TM, Landsat8 | 81.25 (1978) 87.46 (2018) | 0.785 (1978) 0.857 (2018) |
| Proposed Method | Accuracy assessment of Vijayawada, Visakhapatnam, and Tirupati region was performed for the years 2000, 2005, 2010, 2015, and 2020 with the help of Interactive supervised classification | Landsat TM, ETM+, Landsat8 | 97, 95, 92, 92, 90 (2000, 2005, 2010, 2015, 2020-Vijayawada) | 0.96, 0.94, 0.92, 0.9, 0.89 (2000, 2005, 2010, 2015, 2020-Vijayawada) |
| | | | 97, 94.5, 92, 91, 90 (2000, 2005, 2010, 2015, 2020-Visakhapatnam) | 0.96, 0.93, 0.9, 0.89, 0.87 (2000, 2005, 2010, 2015, 2020-Visakhapatnam) |
| | | | 97, 94.6, 92, 92, 91 (2000, 2005, 2010, 2015, 2020—Tirupati) | 0.96, 0.92, 0.91, 0.9, 0.89 (2000, 2005, 2010, 2015, 2020—Tirupati) |

*4.2. LST Analysis*

Land Surface Temperature (LST) was obtained by exploiting remote sensing and GIS methods. Each pixel within the side of the image denotes the surface temperature of each object that can be set by many land-cover forms. Using Mono Window algorithm processing steps, LST maps are created independently for LANDSAT ETM+, LANDSAT 8 information for 2000, 2005, 2010, 2015, and 2020 of 3 areas are proven in Figures 10–12, respectively. The maps confirmed that diverse land-cover kinds have numerous temperature values owed to versions within side the physical traits of the land included with the aid of using the diverse constituents. LST obtained over distinctive classes in Tirupati, Visakhapatnam, and Vijayawada areas is provided in Table 16 and a graphical illustration of obtained LST is represented in Figure 13.

LST acquired over different classes in Tirupati, Visakhapatnam, and Vijayawada regions is shown in Table 16, and a graphical representation of acquired LST is represented in Figure 13, where estimation of LST can be used to interpret the urban development accord on the environment. Sensitivity evaluation at the carried out MW set of rules confirmed that the LST derived from satellite images changed into extra trusty. The extended urbanization sample altered the city's land surface as maximum resistant, favoring the individual increase of LST. It changed into additionally ascertained that the water body, commonly exposes low heat, additionally confirmed an extrude within side the imply temperature over the decennary. The class-wise temperature analysis for the periods: 2000, 2005, 2010, 2015, and 2020, showed that the highest mean temperatures were observed over the built-ups and others, followed by vegetation and water bodies, respectively. This serves as an illustration of urbanization's effects on the environment. The class-sensible temperature evaluation for the periods: 2000, 2005, 2010, 2015, and 2020 confirmed that the best implies

temperatures have been ascertained over the built-ups and others, followed via flora and water bodies, respectively.

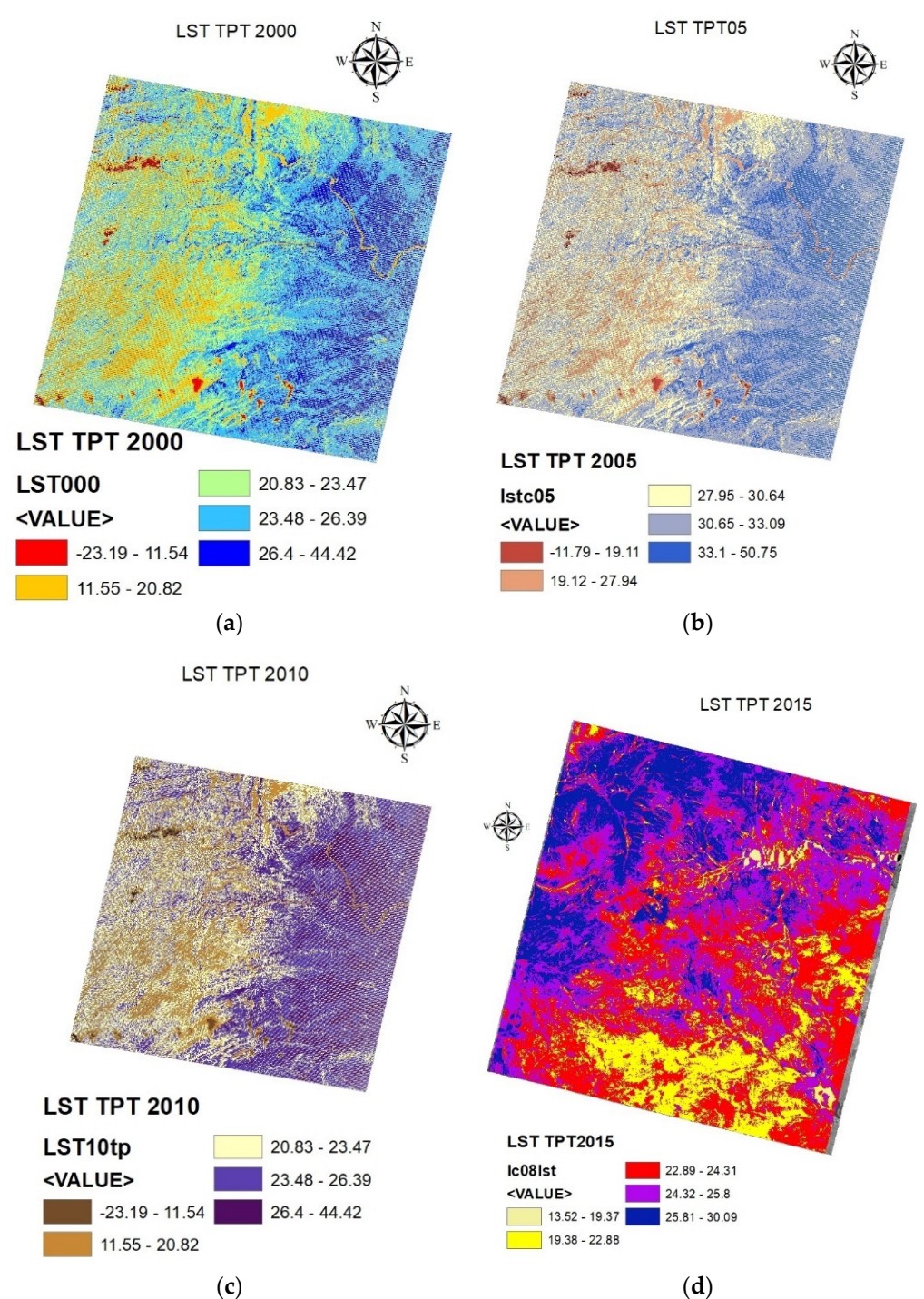

**Figure 10.** *Cont.*

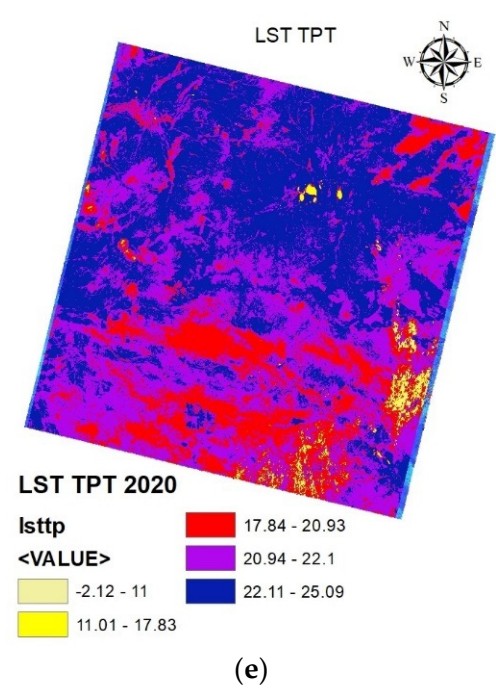

(**e**)

**Figure 10.** (**a**–**e**): LST map of Tirupati region of years 2000, 2005, 2010, 2015, 2020.

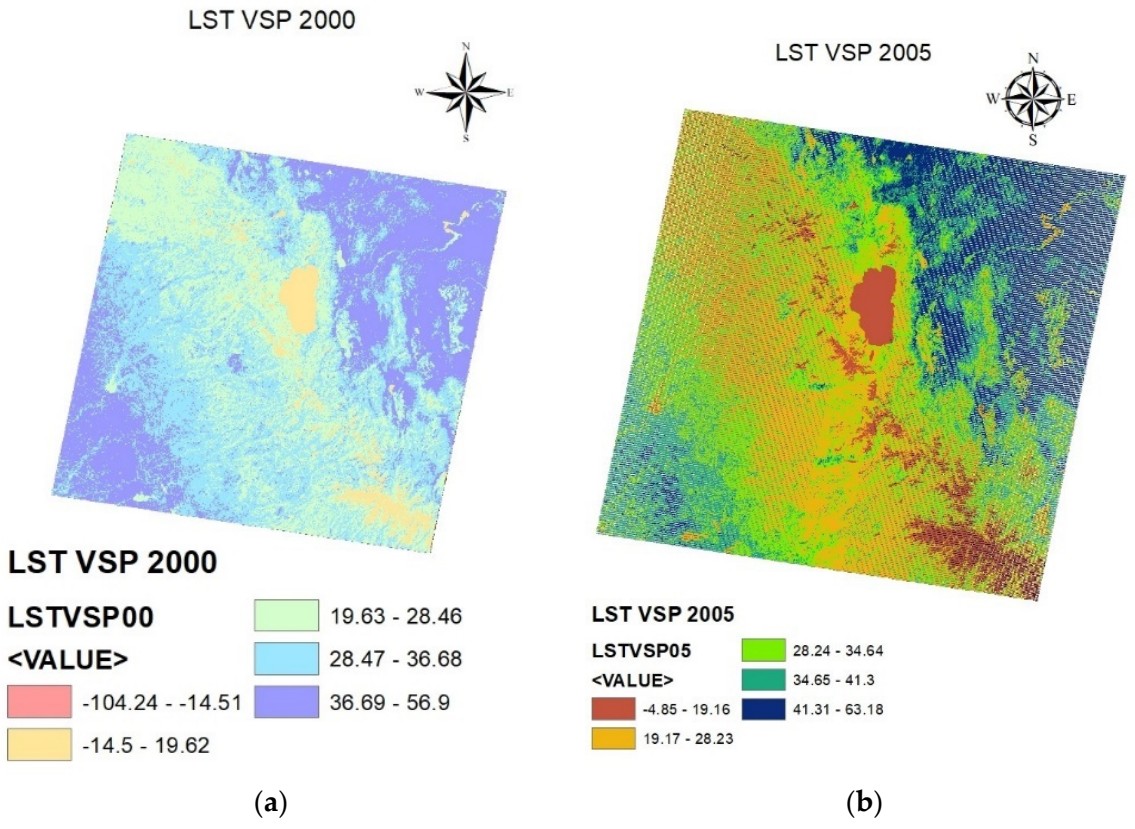

(**a**)                                      (**b**)

**Figure 11.** *Cont.*

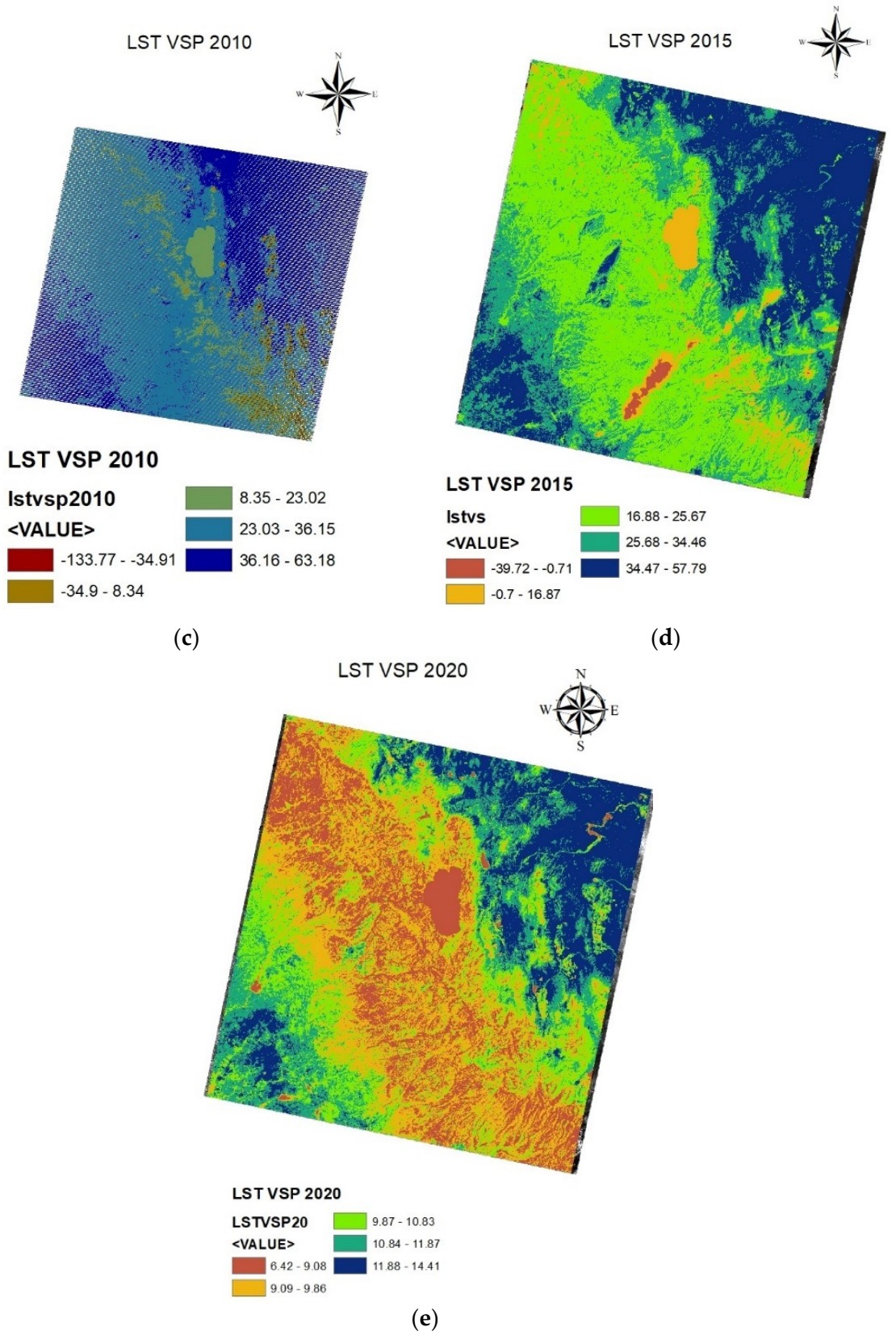

**Figure 11.** (**a**–**e**): LST map of Visakhapatnam region of years 2000, 2005, 2010, 2015, 2020.

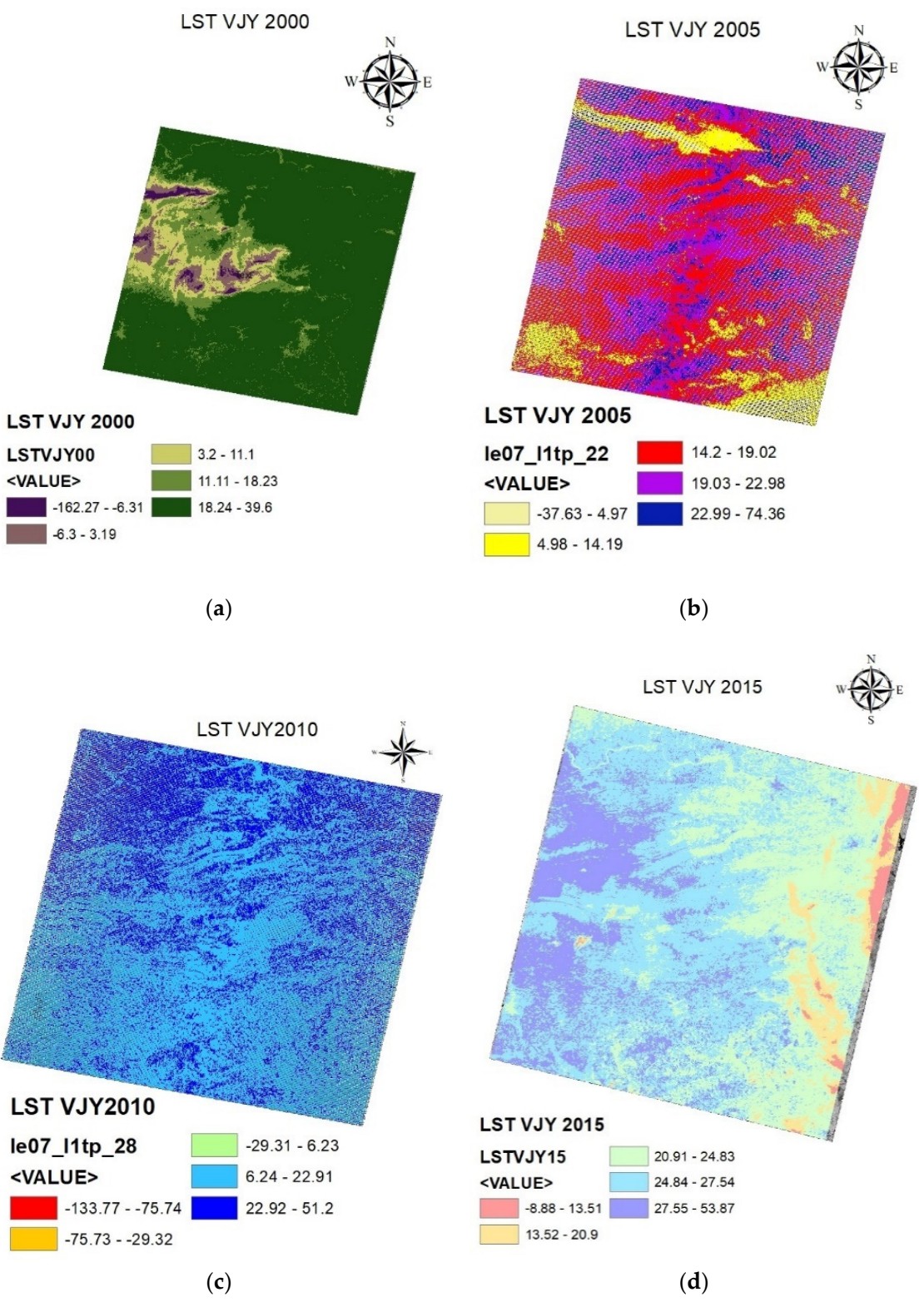

**Figure 12.** *Cont.*

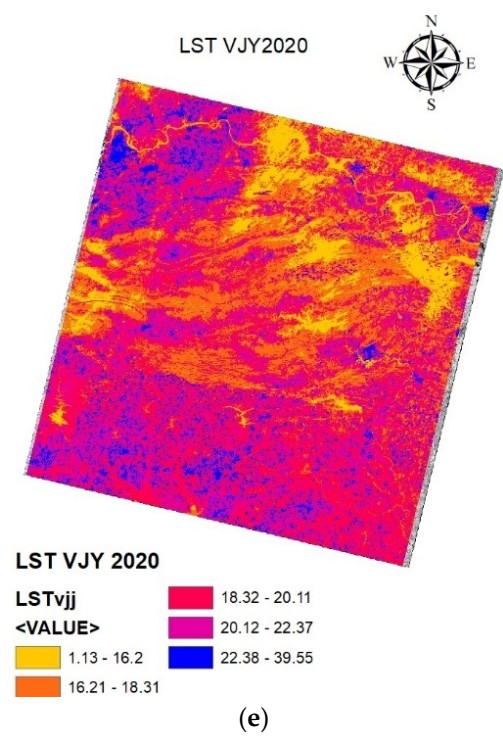

(**e**)

**Figure 12.** (**a**–**e**): LST map of Vijayawada region of the years 2000, 2005, 2010, 2015, 2020.

**Table 16.** LST acquired over different classes in Tirupati, Visakhapatnam, and Vijayawada regions.

| Location | Year | Land-Cover Type | LST Acquired Using the Proposed Method |
|---|---|---|---|
| Tirupati | 2000 | Dense Vegetation | 44.41 |
| | | Vegetation | 26.38 |
| | | Built-up | 23.37 |
| | | Barren Land | 20.82 |
| | | Water | 11.57 |
| | 2005 | Dense Vegetation | 50.74 |
| | | Vegetation | 33.09 |
| | | Built-up | 30.63 |
| | | Barren Land | 27.93 |
| | | Water | 19.11 |
| | 2010 | Dense Vegetation | 44.41 |
| | | Vegetation | 26.38 |
| | | Built-up | 23.47 |
| | | Barren Land | 20.82 |
| | | Water | 11.54 |
| | 2015 | Dense Vegetation | 30.09 |
| | | Vegetation | 25.8 |
| | | Built-up | 24.3 |
| | | Barren Land | 22.877 |
| | | Water | 19.36 |

**Table 16.** *Cont.*

| Location | Year | Land-Cover Type | LST Acquired Using the Proposed Method |
|---|---|---|---|
| | **2020** | Dense Vegetation | 25.08 |
| | | Vegetation | 22.09 |
| | | Built-up | 20.92 |
| | | Barren Land | 17.8 |
| | | Water | 11 |
| **Visakhapatnam** | **2000** | Dense Vegetation | 56.09 |
| | | Vegetation | 36.67 |
| | | Built-up | 28.46 |
| | | Barren Land | 19.61 |
| | | Water | −14.08 |
| | **2005** | Dense Vegetation | 63.18 |
| | | Vegetation | 41.3 |
| | | Built-up | 34.63 |
| | | Barren Land | 28.23 |
| | | Water | 19.16 |
| | **2010** | Dense Vegetation | 63.17 |
| | | Vegetation | 36.14 |
| | | Built-up | 23.01 |
| | | Barren Land | 8.34 |
| | | Water | −34.17 |
| | **2015** | Dense Vegetation | 57.78 |
| | | Vegetation | 34.46 |
| | | Built-up | 25.66 |
| | | Barren Land | 16.87 |
| | | Water | −74 |
| | **2020** | Dense Vegetation | 14.41 |
| | | Vegetation | 11.87 |
| | | Built-up | 10.83 |
| | | Barren Land | 9.86 |
| | | Water | 9.08 |
| **Vijayawada** | **2000** | Dense Vegetation | 39.06 |
| | | Vegetation | 18.22 |
| | | Built-up | 11.1 |
| | | Barren Land | 3.18 |
| | | Water | −6.31 |
| | **2005** | Dense Vegetation | 34.36 |
| | | Vegetation | 22.97 |
| | | Built-up | 19.02 |
| | | Barren Land | 14.19 |
| | | Water | 4.97 |

**Table 16.** *Cont.*

| Location | Year | Land-Cover Type | LST Acquired Using the Proposed Method |
|---|---|---|---|
| | | Dense Vegetation | 51.2 |
| | | Vegetation | 22.91 |
| | **2010** | Built-up | 6.22 |
| | | Barren Land | −29.31 |
| | | Water | −7.51 |
| | | Dense Vegetation | 53.87 |
| | | Vegetation | 27.53 |
| | **2015** | Built-up | 24.83 |
| | | Barren Land | 20.89 |
| | | Water | 13.5 |
| | | Dense Vegetation | 39.54 |
| | | Vegetation | 22.37 |
| | **2020** | Built-up | 20.11 |
| | | Barren Land | 18.3 |
| | | Water | 16.19 |

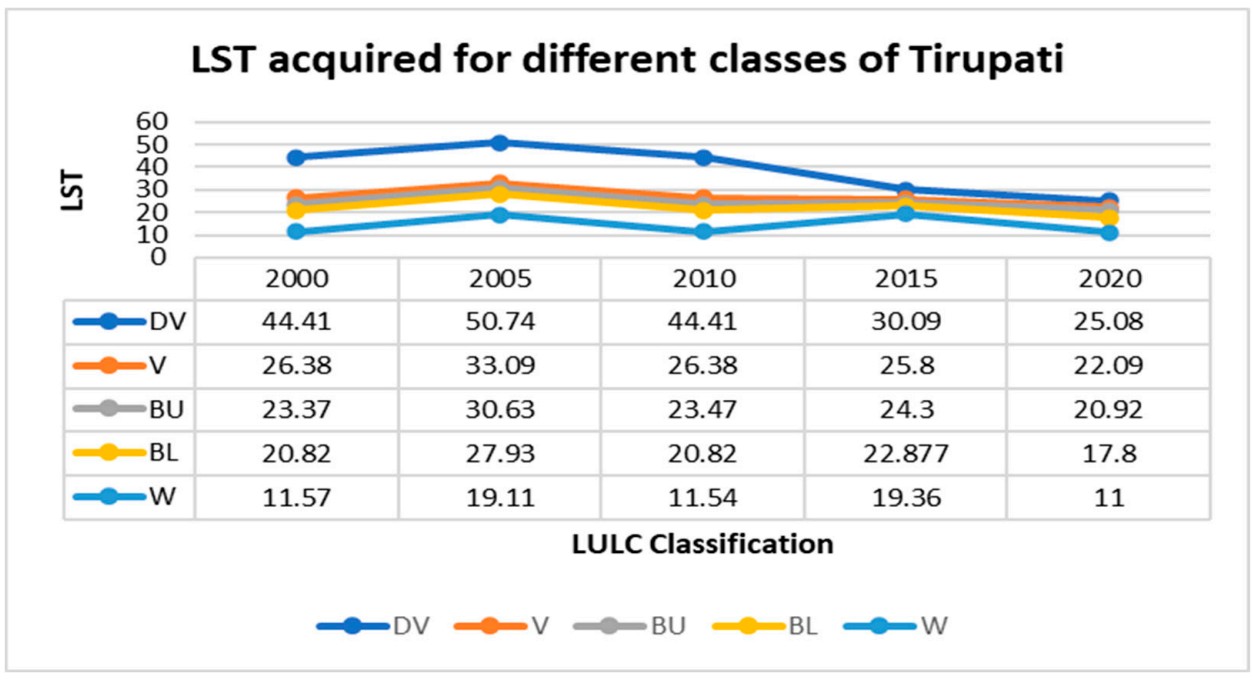

**Figure 13.** *Cont.*

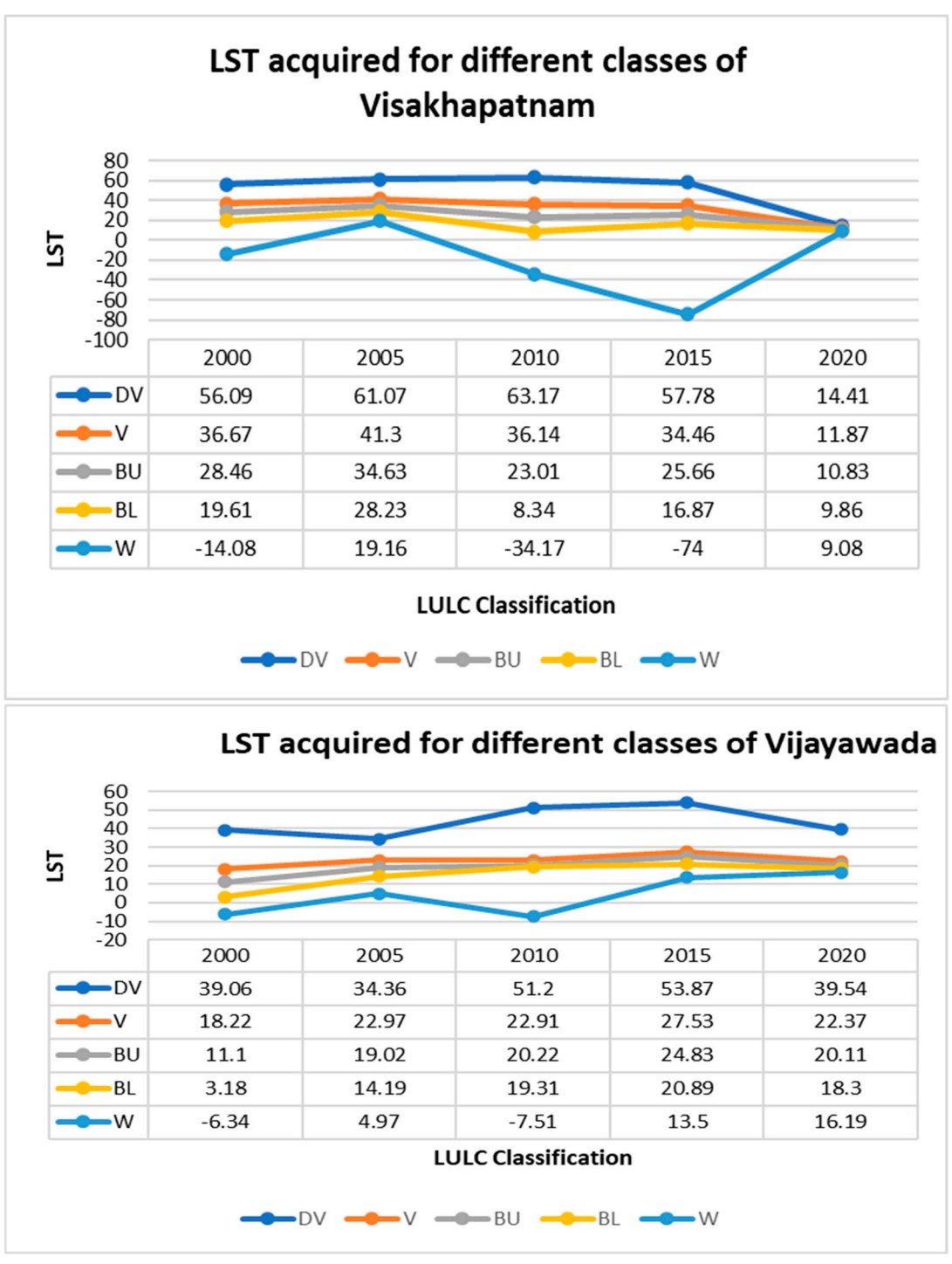

**Figure 13.** LST acquired for different classes of Tirupati, Visakhapatnam, and Vijayawada area.

## 5. Discussion

Visakhapatnam, Vijayawada, and Tirupati cities have perceived unequaled development in the last 2 decades. Along with the urban expansion, these cities have sensed a systematic increase in their temperature. This in turn has exposed the cities to a miscellany of infrastructural and climate-related problems. Urban expansion has come at the cost of agricultural land. Cultivated land has also been acquired in the outskirts of the cities by developers for fancy real estate projects to assist the huge inflow of the population. This has led to increased journey time, a higher number of vehicles on roads, traffic over-crowding, and air pollution from vehicular expel. Pollution is another antagonistic consequence of

the rapid urban expansion in these cities. An increase in surface temperature coupled with the loss of vegetation to the built-up area over the years has made the cities more not defendable to natural hazards, viz. flooding, cyclones, etc. In the present study, visible, near-infrared, and thermal infrared images from the TMLandsat 7 and OLI/TIRS-Landsat 8 sensors were utilized to classify LULC and estimate NDVI and LST. Mono Window Algorithm was used to acquire the LST of images gathered by the sensors throughout the historic series (2000–2020), and LULC classification was performed in the summer of the years 2000, 2005, 2010, 2015, and 2020 in contemplation of verifying the consequences of LULC classes on the temperature from NDVI on a temporal and spatial scale. Seasonally, the highest correlations between LST and area size were verified on the urban area and dense vegetation classes. This outcome points to the dependence of LST values on the existence of vegetation in areas where anthropogenic interference occurred exactly, either through soil water-resistant or the sack of vegetation covers exposing the surface. Thus, human actions qualified the increase in temperature in these areas to neighboring environments, quantifying the phenomenon of urban heat slands (UHIs). The present study found that the removal of vegetation cover and the impacts caused by the LST are of great complexity. LST is considerably influenced by vegetation dynamics. NDVI is widely used to evaluate changes in LST. Therefore, NDVI chose to use it in this work. NDVI can put down a significant amount of the noise caused by atmospheric effects, clouds or cloud shadows, topography, and changing sun angles. However, it is worth noting that it is sensitive to canopy background variations and more saturated at high biomass levels.

## 6. Conclusions

A precise LC map emulates an essential part of conveying modern environmental affairs and agronomy demands. In this research, RS and GIS were unified in measuring and discovering LULC consequences in Tirupati, Visakhapatnam, and Vijayawada regions over a 20-year period from 2000 to 2020. Remote sensing & GIS and satellite-derived images are potent tools, widely exploited in natural asset management. In previous studies, authors used parallelepiped, minimum distance, Mahalanobis, and maximum likelihood classifier methods, obtaining a maximum accuracy of 95% and a kappa coefficient of 0.94. In this research, an interactive supervised class is used wherein there may be no need for signature files, and land cover changes are perceived by exploiting NDVI and PCA for satellite images of five various years: 2000, 2005, 2010, 2015, and 2020, which provides accurate results compared to applying only NDVI. The technique applied in this research is unsophisticated and affordable. The grade of land-use adjustments in the study place changed into one located with the assistance of multitemporal satellite imagery, and class accuracy changed into one calculated by exploiting the confusion matrix. In this 20-year duration, the place below built-up land and vegetation land amassed substantially, while places under agricultural land and water bodies were extraordinarily reduced. LULC outcomes within the side survey place take into consideration the drop-off in agricultural spots and the boom in built-up spots. LST changed into a method received with the aid of using exploiting the mono-window algorithm. The computed LST, and the use of RS and GIS had been authenticated through in situ measures taken over from the identical geographical place. The LULC adjustments will not have a substantial environmental result in the study area. However, the LULC adjustments should be cautiously monitored for the feasibility of the environment. Further, the paintings are probably spread out through exploiting numerous machines and deep learning algorithms that would beautify the algorithm's execution.

**Author Contributions:** Funding acquisition, C.O.S. and T.C.M.; Investigation, V.K.A.S.; Resources, S.L.T. and C.V.; Supervision, D.G. and S.K. All authors have read and agreed to the published version of the manuscript.

**Funding:** This work is supported by School of Electronics and Electrical Engineering, Lovely Professional University, Punjab (India) along with Ministry of Research, Innovation, Digitization from Romania by the National Plan of R&D, Project PN 19 11, Subprogram 1.1. Institutional performance-Projects to finance excellence in RDI, Contract No. 19PFE/30.12.2021 and a grant of the National Center for Hydrogen and Fuel Cells (CNHPC)—Installations and Special Objectives of National Interest (IOSIN). This paper was partially supported by UEFISCDI Romania and MCI through BEIA projects NGI-UAV-AGRO, SOLID-B5G, IPSUS, ADCATER, U-GARDEN, SmartVIT/IoT-NGIN and by European Union's Horizon 2020 research and innovation program under grant agreement No. 883522 (S4ALLCITIES). The work of Chaman Verma was supported by the Faculty of informatics, Eötvös Loránd University, Budapest, Hungary.

**Institutional Review Board Statement:** Not applicable.

**Informed Consent Statement:** Not applicable.

**Data Availability Statement:** Data will be available on reasonable request.

**Conflicts of Interest:** The authors declare no conflict of interest.

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
