# Peer review of "Classification and Validation of Spatio-Temporal Changes in Land Use/Land Cover and Land Surface Temperature of Multitemporal Images"

_sustainability, doi:10.3390/su142315677_

Round 1
Reviewer 1 Report (Previous Reviewer 1)
The paper aims at analyzing LULC changes in three sites using the self-styled classification technique. I would suggest several revisions that should be done before publishing for the Sustainability.
1. From the point of view of the LULC map producing, the classification technique used in this paper is extremely out of date, although the accuracy appears acceptable with the design of authors' experiments. For example, machine learning techniques are widely used recently, obtaining higher and solid accuracies.
2. The sampling design has not been well elaborated in the manuscript. This leads to crucial questions whether the "higher accuracy" obtained by the authors is actually reflecting the classification accuracy! Although the number of samples was mentioned in the manuscript, what type of sampling design supporting the training and testing of authors' classification technique has not been well illustrated!
I then had following questions: 1) Did authors use stratified sampling methods? Or what's the sampling strategy? 2) the sampling design lacks of clarity. As author pointed, the number of samples should be larger than 50 in terms of the rule of thumb. They then used 90-150samples for each LC type. To the best of my knowledge, mapping projects rarely had this rule. I would suggest authors to refer some real LULC mapping project to support their point of determining the number of samples. Any mapping project (e.g., LUCAS?) actually used the rule author taken would be great for the community!
Also, without further descriptions, the "higher accuracy" authors believed actually are not solid for the community.
3. The reason why research the study areas has not been well explained. More descriptions would be great for readers to understand the relationships between your study sites and LULC.
4.The transition matrices displayed in the manuscript looks like a disaster, to me at least, to interrupt the paper. May I suggest move all these tables in the attached files and rearrange them as few tables.
Author Response
Dear Editor,
We have done all the suggested changes and highlighted the same in green in this revision 3.

Reviewer 2 Report (Previous Reviewer 3)
I thank the authors for addressing my questions and comments. However, there are still a few issues that need to be addressed:
1. The authors used several equations to depict the methods used in this study. However, a brief description and explanation of those equations and what they mean is needed. For example, in equation 2, what does X and XT represent? Please add brief description.
2. Please re-make the figures, specially the maps. The current figures do not look very professional. For example, please rename your layers in the legend to make it more meaningful. Also, reduce the number of decimal points to 2 or 3. A very big number after the decimal point such as Figure 10-12 does not mean much.
Author Response
Dear Editor,
We have done all the suggested changes and highlighted the same in green in this revision 3.

This manuscript is a resubmission of an earlier submission. The following is a list of the peer review reports and author responses from that submission.
Round 1
Reviewer 1 Report
Thanks for the revisions.
Unfortunately, I still did do not see enough significant advances in mapping land use/cover in terms of remote sensing and classification techniques. Although the authors stated that they have provided more details on Page 3, they did not really catch the state-of-art of mapping techniques based on remote sensing technology, for example, machine learning methodologies, random forest classification, et al. The elaborations of previous drawbacks of mapping techniques are still extremely far to the state-of-art.
Additionally, the advantages of authors' method still lack of clarity, the real differences from the literature. Moreover, from my point of view, it is rather standard and heavily similar with very outdated methodologies.
I, therefore, unfortunately could not recommend this paper published in the Journal.
Reviewer 2 Report
This study investigated the Spatio-temporal Changes in Land Use/Land Cover and Land Surface Temperature of Multi-temporal Images bye using remote sensing techniques. Remote sensing techniques used in this study are well explained and well concluded. However, there are several issues in present form of manuscript. Specific comments are given below:
1) Introduction should not start with abbreviation. Authors should define the abbreviation, when it appear first time.
2) Quality of most of the figures is not good and presentable. For example text in Figure 1 is not clear. Some figures are streched which need to change. Authors should not change the aspect ratio of original image.
3) Abbreviations used in each table should be well defined in footnotes of table. Need to check for all tables.
4) Each table of result given in the manuscript should be sited in results and discussion part. For example, Table 14 is not sited in results and discussion.
Reviewer 3 Report
Overall, the authors studied land use land cover changes and changes in land surface temperature over time in the study area. However, there are some issues that need to be addressed before the manuscript can be considered further:
1. References need to be properly cited in the manuscript. In many occasions, the authors used the full name of an author in citation which should be last name and year (e.g., Mahmudul Hasan et al. should be Hasan et al. (year of publication)).
2. Section 1.1 - LST change study is almost half of the entire study but the authors did not include that as an objective.
3. The main issue with the manuscript is the Results & Discussion section. I suggest the authors separate this section into two separate section. Results section should include descriptions of their findings. Then an individual Discussion section should include scientific discussions as to what could be the reason behind their findings. Such as, what are/what could be the driving factors behind LULC change and LST change in the study area with a hint of future directions. The current formatting of combining the Results & Discussion section makes it very hard to follow and lacks proper scientific investigations.